# Imaging spatiotemporal evolution of molecules and active sites in zeolite catalyst during methanol-to-olefins reaction

Mingbin Gao [1,2], Hua Li[1], Wenjuan Liu[2,3], Zhaochao Xu[3], Shichao Peng[1,2], Miao Yang[1], Mao Ye [1✉] & Zhongmin Liu [1✉]

Direct visualization of spatiotemporal evolution of molecules and active sites during chemical transformation in individual catalyst crystal will accelerate the intuitive understanding of heterogeneous catalysis. So far, widespread imaging techniques can only provide limited information either with large probe molecules or in model catalyst of large size, which are beyond the interests of industrial catalysis. Herein, we demonstrate a feasible deep data approach via synergy of multiscale reaction-diffusion simulation and super-resolution structured illumination microscopy to illustrate the dynamical evolution of spatiotemporal distributions of gas molecules, carbonaceous species and acid sites in SAPO-34 zeolite crystals of several micrometers that are typically used in industrial methanol-to-olefins process. The profound insights into the inadequate utilization of activated acid sites and rapid deactivation are unveiled. The notable elucidation of molecular reaction-diffusion process at the scale of single catalyst crystal via this approach opens an interesting method for mechanism study in materials synthesis and catalysis.

[1] National Engineering Laboratory for Methanol to Olefins, Dalian National Laboratory for Clean Energy, iChEM (Collaborative Innovation Center of Chemistry for Energy Materials), Dalian Institute of Chemical Physics, Chinese Academy of Sciences, Dalian 116023, People's Republic of China. [2] University of Chinese Academy of Sciences, Beijing 10049, People's Republic of China. [3] Key Laboratory of Separation Science for Analytical Chemistry, Dalian Institute of Chemical Physics, Chinese Academy of Sciences, Dalian 116023, People's Republic of China. ✉email: maoye@dicp.ac.cn; liuzm@dicp.ac.cn

Heterogeneous catalysis has been applied in a wide variety of chemical, pharmaceutical, and environmental processes for more than a century[1–3]. Typically, for the crystalline nanoporous catalysts, owning to the interplay of molecular diffusion[4,5], complex porous structure, and physicochemical properties of active sites, substantial gradients of molecules occur and lead to discrepant stages of chemical reactions at different positions for different time inside even an individual catalyst. Probing dynamic change of molecules and active sites in heterogeneous catalysis, therefore, is vital to understand structure–performance relation critical to product distribution, and deactivation detrimental to catalyst lifetime[1,2,6,7]. As a consequence, directly imaging spatiotemporal evolution of molecules and active sites in individual catalyst of industrial interests is of practical significance.

Progress of various spectroscopic methods in past decades has made it possible to visualize the active sites[2,7], pore structure[6,8], molecular transport and adsorption[5,9,10], chemical transformations[11–15], and heat effect[16] at scale of single catalyst crystal or pellet. For instance, the distribution of Brønsted acidity in a catalyst large than 50 μm was successfully detected by use of synchrotron-based infrared microscopy (IRM)[17]. Highly spatial-resolved observations of acidity were achieved by fluorescence microscopy in conjunction with probe molecules[13,18]. Recent work shows that the collective molecular transport could be

tracked by use of interference microscopy (IFM) and IRM[5,19] in catalysts with crystal size larger than 20 μm[5,9]. The confocal fluorescence microscopy (CFM), which shows extraordinary potential in life science[20,21], was demonstrated capable of capturing mobile trajectories of individual conjugated macromolecule in meso- and macro-pore of fluid catalytic cracking (FCC) catalyst pellets[10], and spatiotemporal evolution of carbonaceous species in model zeolite crystal (larger than 40 μm) in methanol-to-olefins (MTO) and FCC reactions[18,22,23]. Spatiotemporal-resolved evolution of small molecules (e.g., alkene and alkane) and probe molecules-free measurements of acid sites in catalysts of a few micrometers, which are normally encountered in industrial catalytic processes, remain big challenges[2,24].

In current work, we will show a recently proposed deep data approach[25] (as shown in Fig. 1a) can be effectively used to image the spatiotemporal evolution of molecules and acid sites in catalyst of industrial interests. Deep data approach aims to provide physical or chemical insights behind experimental data and fills in the missing information that is hard to obtain via available experimental techniques[25]. In the deep data approach, the advanced theoretical modelling approaches are combined with spatiotemporal-resolved spectroscopy, among other measurements, to obtain the dynamic information of molecules and acid sites in single catalyst[25]. First-principles-based simulations, though

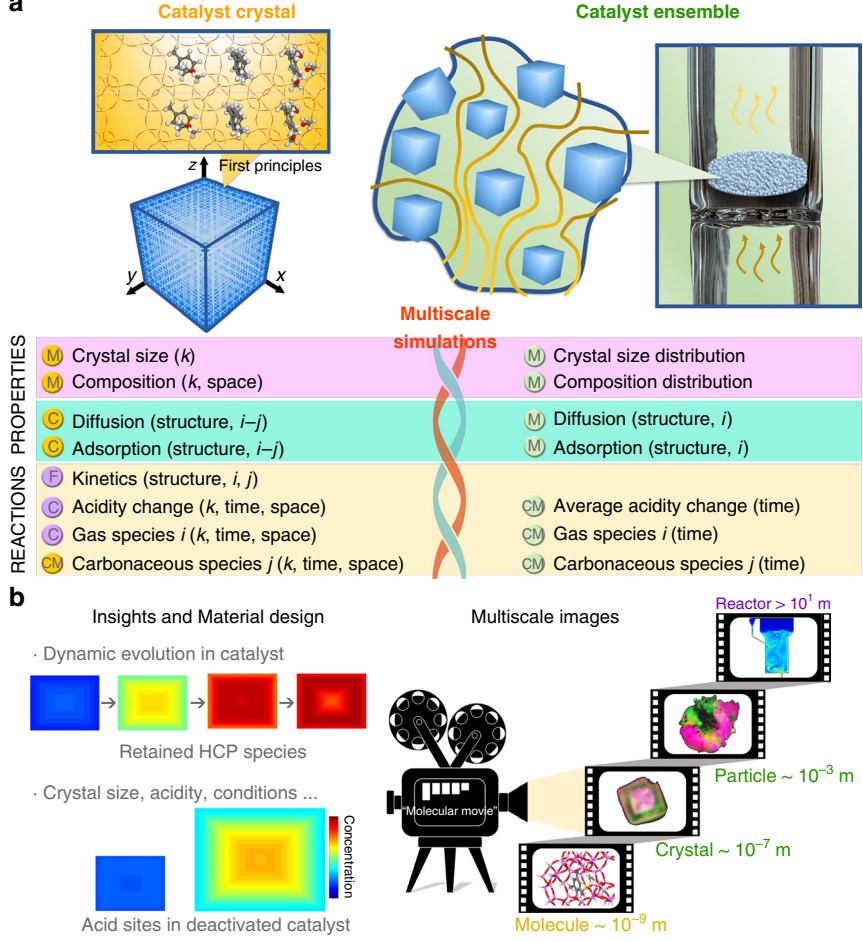

**Fig. 1 Deep data integrates simulations and experiments in application for 'molecular movie'. a** Implementation of the deep data approach to MTO reactions: merging experimental data into multiscale reaction–diffusion modelling. M is measurable, C is calculable, and F is the feedback from experimental data at catalyst ensemble. $i$ is the gas species, $j$ the carbonaceous species and $k$ the individual catalyst crystal. **b** Combination of simulations and experimental methods to unveil the profound insights and implement the 'molecular movie'[2,8] for multiscale of catalytic process. The green and red represented in catalyst crystal or particle are carbonaceous species with low and high molecular weight, respectively. The colour from blue to red represented in reactor is the solid concentration from low to high.

being frequently used in reaction mechanism study, can only reflect the evolution of limited number of reactive sites and trajectories of molecules in a locally minimal region rather than the whole catalyst[22,26]. Instead, a multiscale reaction–diffusion model recently developed for an individual catalyst crystal scale[4,27–29], which integrates the material properties, reaction kinetics and molecular transport, and adsorption[4,27–29], is implemented in this work. This model can provide a link of MTO reaction observed at different length-scales as shown in Fig. 1a. Thus it is possible, by modelling, to comprehend the reaction performance at catalyst ensemble scale on the basis of structures and properties of host materials, evolution of guest molecules, and interplay between host materials and guest molecules at catalyst crystal scale. Spatiotemporal-resolved spectroscopy methods, on the other hand, provide excellent opportunities to examine the rationality of such a model at the scale of single catalyst. Implementation of deep data approach can attain a thorough insight of the evolutions of chemicals and acid sites inside a single catalyst.

Despite its industrial significance[30,31], MTO over zeolite catalysts is a complicated process controlled by sophisticated reaction network and molecular diffusion[32–34]. Carbonaceous species are deposited in the catalyst crystals, leading to the induction, autocatalysis and rapid deactivation. Several methods, for example, solid-state NMR[35,36], dissolution/extraction experiments[33], and ultraviolet–visible (UV–vis) spectra[22,37], have been facilitated to detect carbonaceous species. Nevertheless, at the scale of individual catalyst, the spatiotemporal evolution of carbonaceous species has not been well understood until Weckhuysen et al. first visualized the carbonaceous species in zeolite catalyst for etherification by use of CFM[38]. They observed the carbonaceous species were mainly formed at the rim of crystal during MTO reaction, and speculated that the transformation of diffusion-limited methanol into carbonaceous species preferentially occurs at the rim of crystal[17,39,40]. They further found that co-feeding methanol with water can lead to more homogeneous distribution of carbonaceous species throughout SAPO-34 zeolite crystal[22] since water may alleviate side reactions and enable methanol to diffuse deeply into the crystal. These finding undoubtedly offer essential insights into the spatiotemporal process of deactivation in MTO reaction. However, they employed model zeolite catalysts with large size of 40 μm due to the constraints of spatial resolution of CFM[12,17,40,41]. In MTO industrial catalysts, SAPO-34 zeolite crystals of a few microns are typically used, which are, however, out of the measurement scope of CFM. Direct imaging the evolution of molecules and acid sites in catalysts of a few microns, which is highly desired to understand the mechanism underlying MTO process, therefore, is still a non-trivial task.

To tackle the aforementioned challenges, in this work, the deep data approach is implemented to visualize the evolution of gas molecules, carbonaceous species and acid sites in MTO reaction over SAPO-34 zeolites. Taking full advantages of this approach, the profound molecular reaction–diffusion mechanism in MTO reaction can be unveiled. Essentially, two major puzzles for MTO reaction can be well understood, i.e., the inadequate utilization of acidity and activated species, and rapid catalyst deactivation, which demonstrates that the deep data approach can be effectively used to understand the heterogeneous catalysis process at the scale of single catalysts as shown in Fig. 1b.

## Results

### Implement of deep data approach to MTO reactions.
The idea underlying deep data follows Kalinin et al.[25]. It is to merge elaborative experimental analysis into an established theoretical model, first to estimate the input parameters and validate the model, and then to provide further predictions at the length-scale

concerned. In the deep data approach considered in this work (Fig. 1a), a multiscale reaction–diffusion model (see Supplementary Equations 1–10) provides a link of MTO reaction observed at different length-scales. In the model, the change of molecular loading is described by reaction kinetics and molecular flux via the reaction–diffusion equation (Supplementary Equation 1). The reaction kinetics is developed based on the dual-cycle mechanism[33]. The molecular flux is calculated by Maxwell–Stefan equation (Supplementary Equation 2) and ideal absorbed solution theory[29], which needs the input of molecular diffusivities, Langmuir adsorption parameters, and adsorption enthalpy of the zeolitic framework. In addition, the crystal size and quantity of acid sites are two fundamental parameters required to solve the reaction–diffusion equation for individual catalyst crystal. The molecular flux at the catalyst surface is then used to connect the mass transfer between crystal and gas phase. Given that the properties of individual catalyst crystals and molecular flux at the crystal surface are known, the catalyst ensemble will be simulated by well-established fixed-bed model (Supplementary Equation 10), which the effects of space velocity and partial pressure of methanol can be considered. In this way, the deep data approach is expected to shed lights on the mechanism of MTO reactions supposed that some key parameters at catalyst crystal scale such as guest molecular diffusivities, adsorption isotherms, crystal size, quantity of acid sites, and rate constants of reaction kinetics can be carefully obtained. Therefore, it requires extreme care in deriving the experimental data and analyzing the systematic errors in the model[25]. First of all, preparing SAPO-34 zeolite crystal sample with a relatively uniform distribution of crystal size and acid sites is crucial for the simulations and measurements. The CHA topologies and micropore structures of SAPO-34 samples are also premise for molecular dynamics (MD) simulations, which were carefully tested and shown in Supplementary Figs. 1 and 4. The molecular diffusivities and adsorption isotherms were obtained either from MD or uptake experiments[42]. For all SAPO-34 zeolite samples, the maximum deviation of 15% was achieved for crystal size and 6% for silica content, as shown in Supplementary Table 1. Note that ~0.1 silica content corresponds 1 mmol·$g_{zeo.}^{-1}$ acid sites (Supplementary Table 1), the quantity of acid sites for individual crystal of samples was estimated to be 1.00 ± 0.06 mmol·$g_{zeo.}^{-1}$. The parameters of reaction kinetics, as it is hard to directly obtain at single zeolite scale, were obtained by fitting modelling results with experimental data at catalyst ensemble scale, in terms of catalyst lifetime, product distributions and evolution of acid sites and carbonaceous species on bulk SAPO-34. For MTO reactions over catalyst ensemble, the errors of WHSV and reaction temperature were controlled within 2 and 0.2%. As shown in Supplementary Table 4, the systematic errors of simulations caused by distribution of crystal size and quantity of acid sites, as well as error of WHSV, were also carefully examined. The simulated spatiotemporal distribution of carbonaceous species at zeolite catalyst crystal scale was validated with the advent of SIM. With the validated modelling, the deep data approach can fill in the missing information of MTO reactions at concerned length-scale and provide insights to MTO reaction catalyzed by SAPO-34 zeolites.

### Spatiotemporal evolution of carbonaceous species in SAPO-34.
For simplification, we classified the carbonaceous species as HCP species and coke precursors based on the reactivity and the role on molecular diffusion of carbonaceous species. According to the density functional theory (DFT)[43,44] and experimental results[41,45,46], the methylbenzenes and methylnaphthalene retained in SAPO-34 zeolites can serve as activated carbonaceous species[46,47]. In contrast, the phenanthrene, pyrene or carbonaceous species with

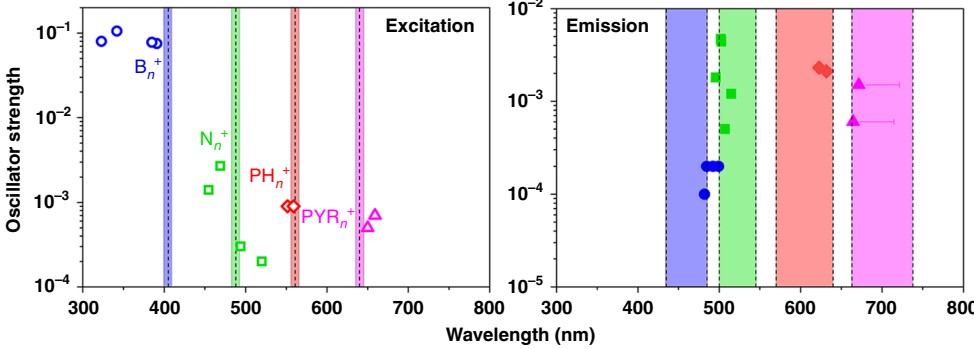

**Fig. 2 Calculated excitation and emission wavelengths of charged carbonaceous species.** Simulated excitation (first excitation energies)[49,50], emission (excited states) wavelengths, and oscillator strength of charged carbonaceous species in gas phase calculated at the B3LYP/6-31G (d, p) level of theory (see also Supplementary Table 5). The lines (or bands) around 405 (435–485 nm), 488 (500–545 nm), 561 (570–640 nm), and 640 nm (663–738 nm) used as wavelengths of excitation (emission detection) by SIM are indicated in blue, green, red, and pink, respectively. $B_n^+$, $N_n^+$, $PH_n^+$, and $PYR_n^+$ stand for benzenic, naphthalenic, phenanthrenic, and pyrenic carbocation with $n$ methyl substituents, respectively.

higher molecular weight (MW)[48] exhibit very low reactivity[44,46]. Our MD simulations show that the loading of methylbenzenes and methylnaphthalene in CHA cage has only slightly affected the molecular diffusion, i.e., the ratio of diffusivity in cage loading with carbonaceous species $D_{load}$ to that of empty cage $D_{empty}$ is 0.3~1. The ratio of $D_{load}/D_{empty}$ dramatically decreases to 0.05~0.1 if loaded with phenanthrene (Supplementary Figs. 15–16), which means that bulky carbonaceous species could significantly limit molecular diffusion in CHA. Thus, the methylbenzenes and methylnaphthalene are classified as HCP species, while phenanthrene, pyrene, and carbonaceous species with higher MW are considered to be coke precursors[41,46].

To directly visualize the spatiotemporal evolution of carbonaceous species inside SAPO-34 zeolite crystals, the multiscale reaction–diffusion simulations and super-resolution SIM measurements were performed in parallel for MTO reaction. As shown in Fig. 2 and Supplementary Table 5, by use of time-dependent density functional theory (TDDFT) calculations[49,50], the calculated excitation wavelengths of benzenic ($B_n^+$), naphthalenic ($N_n^+$), phenanthrenic ($PH_n^+$), and pyrenic ($PYR_n^+$) carbocations with $n$ methyl substituents are situated around 390, 480, 560, and 640 nm, respectively, and the corresponding emission wavelengths are located in the range of 480–490, 500–520, 620–630, and 670–700 nm, respectively. Compared with that of fluorescence signal, the wavelength of phosphorescence signal for given species is about 80 nm higher[51]. The sufficiently close wavelengths of illumination and emission detection of SIM were used in this work, which can cover the characteristic area of excitation and emission wavelengths of $B_n^+$, $N_n^+$, $PH_n^+$, and $PYR_n^+$, and could neglect the interference of phosphorescence. As shown in Fig. 2, it is feasible to quantitatively illustrate spatial distribution of a given type of carbonaceous, i.e., HCP species ($B_n^+$ and $N_n^+$) and coke precursors ($PH_n^+$ and $PYR_n^+$), excited at a certain wavelength.

As in Fig. 3, it can be noted that the peculiar distribution of carbonaceous species was observed in smaller SAPO-34 zeolite crystals. By taking a series of images of, for instance, the SAPO-34-12 sample, one might find that at the initial stage the formation of carbonaceous species starts from the center of crystals, which subsequently expands from the center to the rim. As MTO proceeds, HCP species begin to evolve to coke precursors at the crystal center. Correspondingly, the fluorescence intensities of HCP species at the center are lower than that at the surrounding regime. When the catalyst approaches deactivation, coke precursors are more concentrated at the rim of crystal. The spatial distribution of carbonaceous species exhibits that coke

precursors at the rim surround the HCP species, and HCP species encircle the coke precursors located at the center of crystal. Further reducing the crystal size, as shown in Fig. 3a, for SAPO-34-5 sample, leads to the similar patterns of spatial distribution of carbonaceous species. After catalyst deactivation, however, the distribution of both HCP species and coke precursors in SAPO-34-5 sample becomes more uniform than that in SAPO-34-12. If the crystal size increases, it is found that the spatiotemporal evolution of carbonaceous species in large crystal manifests big difference. As can be seen in Fig. 3f, the formation of carbonaceous species for SAPO-34-50 sample starts from the rim of crystal. With MTO reaction proceeding, the distribution of carbonaceous species gradually expands to the interior of crystal[17,22]. After catalyst deactivation, the fluorescence intensities at the center become quite weak, implying that the carbonaceous species are hardly formed at the center of large crystals.

In Fig. 3, the distributions of HCP species and coke precursors obtained from multiscale reaction–diffusion simulations qualitatively agree well with the SIM images for SAPO-34 zeolites with different crystal size. Figure 3 also shows the standard error of simulations, which might be caused by the distribution of crystal size and acid sites (Supplementary Table 1) and/or error in estimating WHSV. In particular, the spatiotemporal evolution of carbonaceous species in small crystals (e.g., SAPO-34-5, SAPO-34-12, and SAPO-34-17 samples) which are difficult to visualize by CFM[17,39], are clearly imaged. As can be seen, after catalyst deactivation, HCP species trapped inside the crystals are difficult to be accessed and the corresponding spatial distribution shows to be in the shape of 'annulus'. Based on the simulations, the quantity of HCP species in this 'annulus' region increases as the crystal size increases, meanwhile the quantity of coke precursors formed at this 'annular' region decreases (Fig. 3c, d, g). This implies that, for smaller crystals, HCP species inside the crystals can be adequately transformed into coke precursors before catalyst deactivation. The SIM and simulated images revealed that the spatiotemporal evolution of carbonaceous species is decisively controlled by crystal size, which has not been observed in the previous investigations due to the limited spatial resolution[8,17,39,40].

MTO reaction catalyzed by SAPO-34 zeolite with high silica content as an extended study case to illustrate that the deep data approach be potentially used to predict the influence of catalyst properties on MTO reaction. As shown in Supplementary Fig. 6, the influence of silica content of SAPO-34 zeolites on catalyst lifetime, product distribution, changes of acidity and spatiotemporal evolution of carbonaceous species can be simulated, which agree well with the experimental results. By use of the multiscale

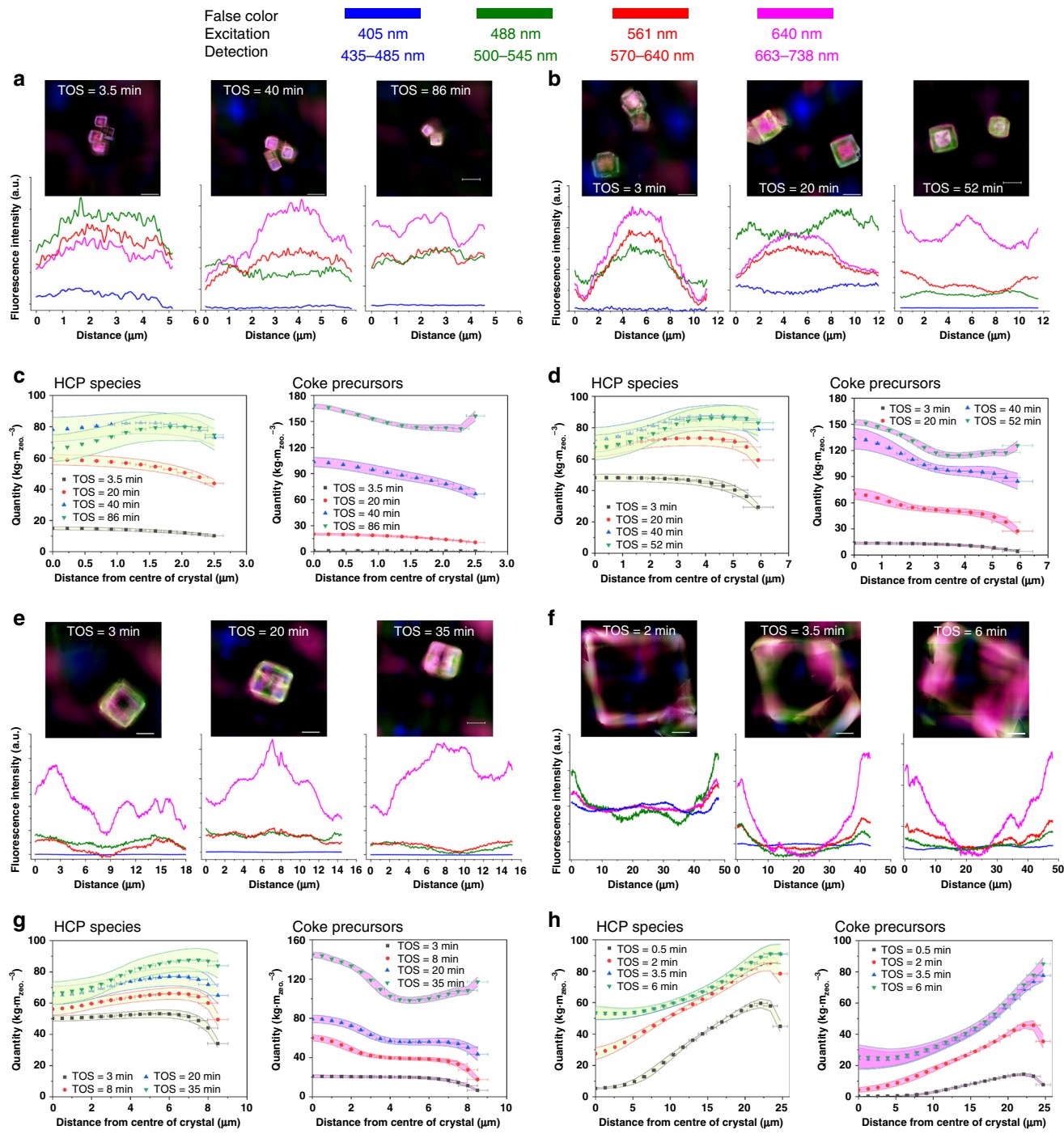

**Fig. 3 SIM images and multiscale reaction–diffusion simulations of an SAPO-34 zeolite crystal.** The spatiotemporal distribution of carbonaceous species obtained from SIM and simulations in **a**, **c** SAPO-34-5 (4.82 ± 0.36 μm), **b**, **d** SAPO-34-12 (11.17 ± 1.80 μm), **e**, **g** SAPO-34-17 (16.92 ± 1.66 μm), **f**, **h** SAPO-34-50 (47.08 ± 3.50 μm) samples during MTO reactions at 723 K and *WHSV* of 5.0 ± 0.1 $g_{MeOH} \cdot g_{zeo.}^{-1} \cdot h^{-1}$. The fluorescence intensities along selected line are also displayed. The quantity of acid sites is 1.00 ± 0.06 $g_{zeo.}^{-1}$. The error band is the standard error of simulation results. The SIM images shown is the fluorescence that originated from the overlap of four profiles with a laser excitation of 405 nm (detection at 435–485 nm, false color: blue), 488 nm (detection at 500–545 nm, false color: green), 561 nm (detection at 570–640 nm, false color: red), 640 nm (detection at 663–738 nm, false color: pink). The images were taken in the middle plane of the zeolite crystal. The scale bar represents 10 μm.

reaction–diffusion simulations, the insights of insufficient utilization of acid sites and HCP species during MTO reaction catalyzed by SAPO-34 with high silica content are also unveiled.

**Spatiotemporal evolution of gas molecules and acid sites in SAPO-34.** In Fig. 4, the spatiotemporal distribution of gas

molecules and acid sites in the SAPO-34-5 sample is presented. At the early stage of MTO reaction (TOS = 3 min), the concentration of methanol enriches at the rim of crystal, owning to the constraint of intracrystalline diffusion by nanopore[5]. Thus, the consumption rate of methanol at the rim is consequently increased because of the enriched concentration (Supplementary Fig. 18a), and the formation rate of gas products in turn becomes

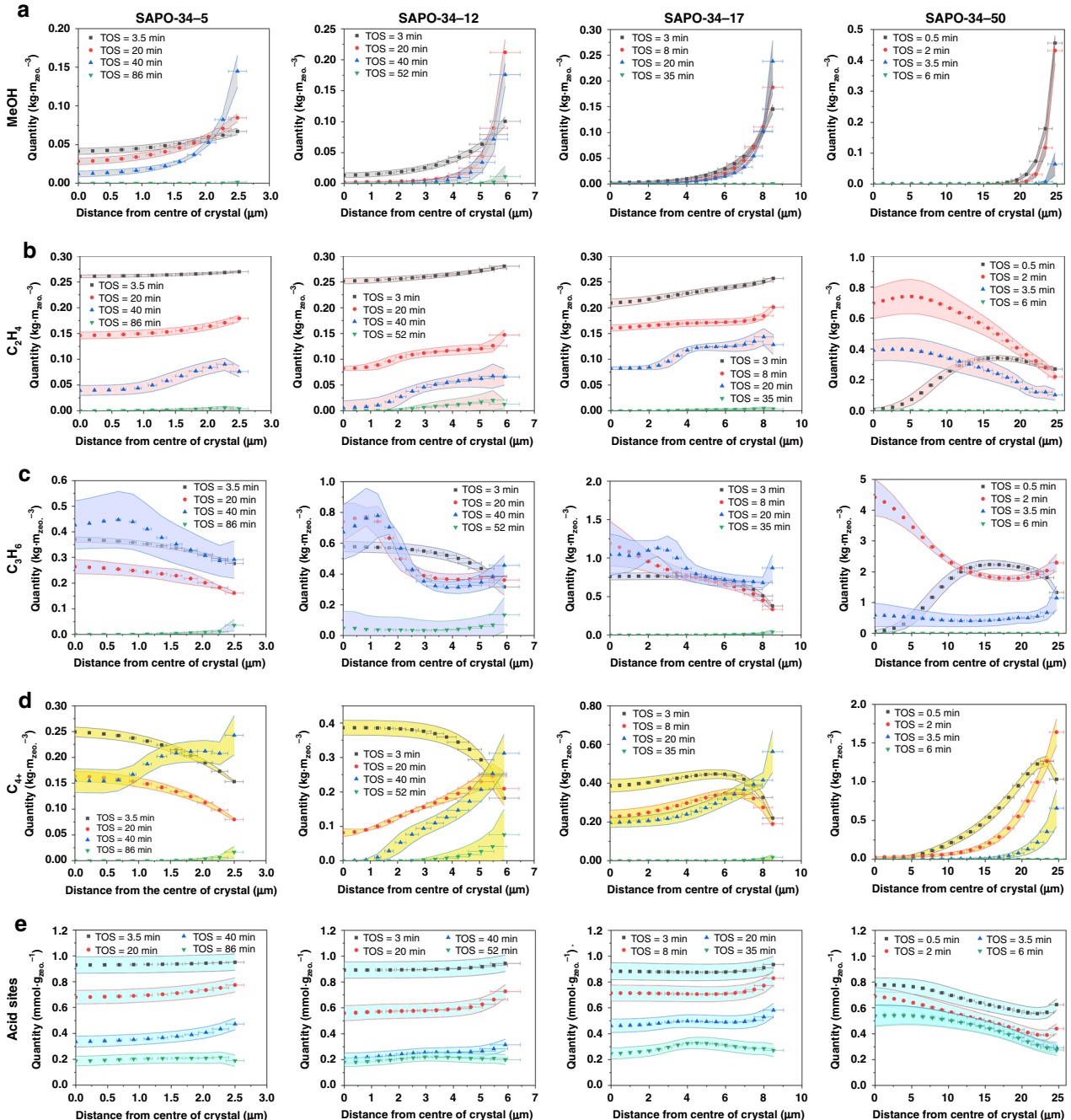

**Fig. 4 Spatiotemporal evolution of gas molecules and acid sites in SAPO-34 zeolite crystals.** Simulated spatiotemporal evolution of concentration of **a** methanol, **b** ethylene, **c** propylene, **d** $C_{4+}$, and **e** acid sites in SAPO-34-5, SAPO-34-12, SAPO-34-17, and SAPO-34-50 crystals during MTO reactions at 723 K and WHSV of $5.0 \pm 0.1$ $g_{MeOH}·g_{zeo.}^{-1}·h^{-1}$. The quantity of acid sites is $1.00 \pm 0.06$ $mmol·g_{zeo.}^{-1}$. The results obtained from multiscale reaction–diffusion simulations. The error band is the standard error of simulation results.

higher. However, the high diffusion flux of gas products at the rim leads to relatively low gradients of concentrations of gas products (Supplementary Fig. 19). As shown in Fig. 4c, d, methanol can easily get access to the center of crystals due to the short diffusion length, and the concentration of propylene and $C_{4+}$, which is formed from methanol, is higher at the center. On the other hand, the low diffusion flux of gas products at the center (Supplementary Fig. 19) makes the gas products accumulated at the center and further transformed to HCP species by cyclization reaction[52]. Correspondingly, the coverage of acid sites begins from the center of crystal (Fig. 4e). In Fig. 4b–d, it can be even distinguished that the difference of the diffusivities of gas

products causes more $C_{4+}$ than propylene can be accumulated at the center. Ethylene normally has high diffusivity and can diffuse out of the crystal smoothly. But at this stage, the propylene is dominantly formed by olefins-based cycle, therefore, the selectivity of propylene is still higher than that of ethylene. In the process of gas products at the center diffusing outward the crystal, it is normally accompanied with cyclization reactions and thus the formation of HCP species gradually expands from at the center to the rim of the crystal. When the MTO reaction proceeds (TOS = 9 min, as shown in Fig. 4a), the increased concentrations of HCP species and coke precursors throughout the crystal, suppresses methanol to diffuse into the crystal center. Thus, the

gradient of methanol concentration at the rim increases. If TOS further increases (TOS = 40 min), the cumulative HCP species at the center of crystal react with propylene and $C_{4+}$ and form more coke precursors[44,53] (Supplementary Fig. 18c). Therefore, the distribution pattern of HCP species turns into the shape of 'annulus' with low concentration of HCP species at the center. More coke precursors meantime formed at the crystal center will significantly limit gas molecules to access the center (Supplementary Fig. 18) and the cumulative coke precursors at the rim restrict gas products to diffuse outward crystal. Hence, as shown in Fig. 4c–e, the patterns of gas products and acid sites also turn into the shape of 'annulus'. Noteworthy phenomenon in Fig. 4c–e is that the concentration of detained propylene and $C_{4+}$ remarkably increases. MD simulations indicate that the intra-crystalline diffusivity of ethylene is still higher than that of propylene in presence of coke precursors (Supplementary Fig. 16). As shown in Fig. 4b, the ethylene with higher diffusion flux (Supplementary Fig. 19a) is mainly distributed close to the rim of crystal. This is responsible for the increased selectivity of ethylene at the beginning of catalyst deactivation (Supplementary Fig. 5). As the gradient of methanol concentration at the rim of crystal increases, HCP species tend to react with methanol rapidly to form coke precursors. A large quantity of coke precursors accumulated at the rim hinder the diffusion of methanol, resulting in the dramatically decrease in methanol conversion. In addition, as shown in Fig. 4c, d, though propylene and $C_{4+}$ are almost completely trapped inside the crystal, the further reactions with HCP species to form coke precursors still occur[33,54]. This can explain that, even the methanol conversion significantly declines or even cutting off the inlet flow, the formation of coke precursors still continues for a while[33,46].

For SAPO-34-50 sample, due to the prolonged diffusion pathway, methanol can only reach at the region closed to the rim when TOS = 0.5 min. In such a short time, the reaction rate of methanol in SAPO-34-50 sample can be four times higher than that in SAPO-34-5 sample (Supplementary Fig. 18a). In Fig. 4b–d, the formation of gas products starts from the region close to rim (~10 μm from the rim). Compared with results of SAPO-34-5, the extended diffusion pathway causes a large quantity of ethylene retained inside the crystal, which results in the decreased selectivity of ethylene (Supplementary Fig. 5d). In Fig. 4e, the formation of HCP species preferentially occurs at the rim and the coverage of acid sites starts from the region close to the rim. When TOS increases to 2 min, a part of gas products generated at the rim diffuses inward center of crystal since the concentration of products is relatively low there (Supplementary Fig. 19). Then, gas products at the center are transformed into HCP species because of the high concentration of acid sites at the center of crystal. Meanwhile, due to the increased gradient of methanol concentration (Fig. 4a), methanol and HCP species located close to the rim can be transformed into respectively HCP species and coke precursors at much higher reaction rates (Supplementary Fig. 18b, c). When TOS = 3.5 min, the occupation of coke precursors at the region close to crystal rim (~2 μm from the rim) significantly hinders the accessibility of HCP species to methanol, leaving only HCP species at the outmost rim continues to react with methanol. Under such circumstance, a large quantity of acid sites are unexploited at the crystal center (Fig. 4e), which is accordance with the results by synchrotron-based IRM[17]. The spatiotemporal evolution of gas molecules and acid sites in the SAPO-34-12 and 17 samples are shown in Fig. 4. As can be seen, at the initial stage, prolonging diffusion length limits the formation of HCP species at the region close to rim (~6 μm from the rim) for the SAPO-34-17 sample. But unlike in the SAPO-34-50 sample, the gas products formed near the rim in the SAPO-34-17 sample (Supplementary Fig. 19) can rapidly access

to the center of crystal and form higher concentration at the center due to the shorter diffusion length. Then the propylene and $C_{4+}$ at the center are transformed to HCP species and further to coke precursors with higher reaction rate compared with that of small crystal (Supplementary Fig. 18b). As a consequence, the formation of HCP species starts from the rim of crystal while coke precursors from the center in SAPO-34-17 as observed in Fig. 3e.

Figure 5 summarizes the role of crystal size, i.e., diffusion length, in determining the accessibility of acid sites, evolutions of carbonaceous species, diffusion of molecules during MTO reactions. Shorting the diffusion length leads methanol to readily diffuse into the center of crystal and thus fully access to the acid sites inside crystal. In addition, gas products formed inside crystal tends to diffuse toward the rim of crystal rather than the center of crystal. Combining with multiscale reaction–diffusion simulations, the evolution of gas molecules, acid sites and carbonaceous species in the zeolite crystals of a few microns can be visualized, which are not possible with existing measurement techniques[2,7] or first-principles-based simulations solely.

**Interpretation of macroscopic phenomena by microscopic image.** Visualization of gas molecules, carbonaceous species and acid sites in SAPO-34 zeolite crystals during MTO reaction, as shown in Figs. 3 and 4, can offer a direct way to unveil the mechanisms underlying, for instance, different utilization of acid sites, discrepant nature and quantity of carbonaceous species, rapid deactivation of catalyst, and change of catalyst lifetime with different crystal size observed macroscopically. In Fig. 6b, DRIFT spectra show that decreasing the crystal size can cause a reduced coverage rate and decreased number of residual acid sites inside SAPO-34 zeolite. It further leads gas molecules to favorably diffuse toward the rim of crystal, which in turn enhances the accessibility of acid sites and alleviates formation of carbonaceous species. In this connection, the transformation rate of HCP species to coke precursors is also decreased (Supplementary Fig. 18c). In Fig. 6e, DR UV–vis spectra provide semiquantitative comparison of a given type of carbonaceous species excited at the same wavelength for different SAPO-34 zeolite samples at different MTO reaction stages. It is difficult to directly correlate the results of UV–vis spectra and SIM images owning to the different absorbed and emissive responses of a given carbonaceous species excited at the same wavelengths as in Fig. 2. From DR UV–vis spectra in Fig. 6e, compared with spectra of large crystal, it can be found that at the initial stage, the relative established rate of $B_n^+$ (~390 nm) and $N_n^+$ (~480 nm) is sluggish in smaller crystals. In addition, the adsorption bands at ~561 nm ($PH_n^+$) and ~640 nm ($PYR_n^+$) are relatively unconspicuous in the small crystal. It has been recently shown that carbonaceous species located in adjacent cavities can be further connected through windows during reaction to form heavier carbonaceous species[48,55]. As can be seen from Fig. 4a, in the small SAPO-34 zeolite crystal, the concentration gradient of methanol through crystal is relatively uniform. This reduces the formation rate of coke precursors at the rim, which significantly promotes the accessibility of HCP species to gas molecules and therefore improves the transformation of HCP species to heavier coke precursors. Thus, in the smaller zeolite crystals, the quantity of retained carbonaceous species and the MW of coke precursors both increase, which can be reflected by TGA (Fig. 6a), DR UV–vis spectra (i.e., bands at ~561 and ~640 nm become stronger and wider after deactivation, Fig. 6e), GC-MS (Fig. 6c), and MALDI FT-ICR MS (Fig. 6d).

Rapid deactivation of zeolites in MTO reaction, despite the significance for industrial applications[8,56], is still an unresolved scientific puzzle. In Fig. 5a, the formation of coke precursors at

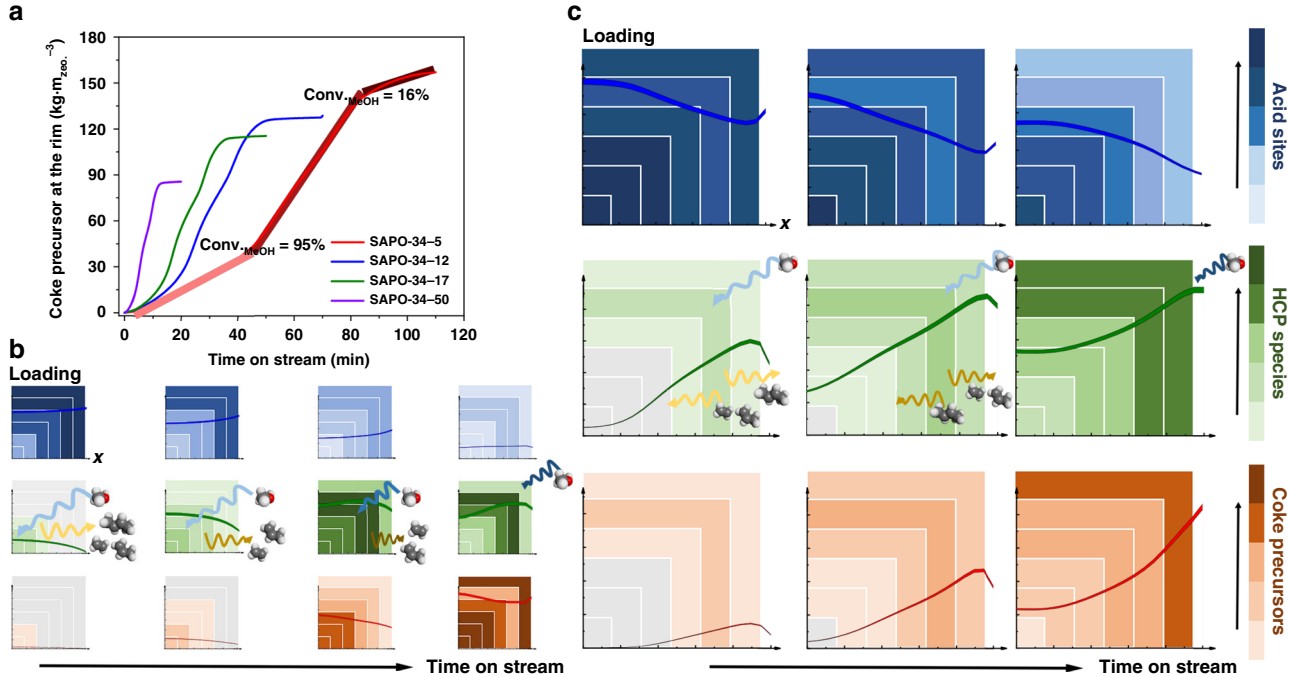

**Fig. 5 Molecular reaction–diffusion mechanism during MTO reaction at the zeolite crystal level. a** The evolution of coke precursors at the rim of SAPO-34 zeolite crystals with different crystal size. **b**, **c** Schematics of spatiotemporal evolution of acid sites, HCP species and coke precursors during MTO reaction in small and large crystal size, respectively. 2D plots represent the X–Y plane of center section of crystal, which starts from the centre of crystal. The curves present the loading of acid sites and carbonaceous species as the function of the distance from the centre to edge of the crystal.

the rim of crystal can be divided into three stages. At the first stage, i.e., the methanol conversion is higher than 95%, the formation rate of coke precursors at the rim of crystals in SAPO-34-5 sample is about 0.05 times of that in SAPO-34-50 sample (Supplementary Fig. 18c). This favors the diffusion of methanol in zeolite crystal and the accumulation and transformation of HCP species, which can effectively prolong the duration of high methanol conversion in smaller crystals. At the second stage, the formation rate of coke precursors by transformation of HCP species increases by more than two times because the methanol is significantly diffusion-limited. The accelerated pore blockage at the rim by coke precursors further limits the diffusion and reaction of methanol. Interestingly, this stage can be directly associated with the rapid descent of methanol conversion, as in Fig. 6a. This means quantitatively unveiling the mechanism of rapid deactivation by current approach is possible. Figure 4e shows that the quantity of residual acid sites at the rim of crystal in SAPO-34-5 is significantly less than that in SAPO-34-50. This implies that the capacity of coke precursors at the rim of smaller crystals enlarges, as the rate of methanol conversion drops from 95 to 20% decreases as shown in Fig. 6a. Thus, the maximum quantity of coke precursors at the rim may increase if the crystal size decreases (Fig. 5a), which explains the darker color of deactivated SAPO-34 zeolite crystals with smaller size (Fig. 6e).

## Discussion
In this work, we synergistically combine the multiscale reaction–diffusion simulations with spatiotemporal-resolved SIM spectroscopy, among other measurement techniques, via a deep data approach, to visualize the spatiotemporal evolution of gas molecules, carbonaceous species and acid sites in MTO reaction over SAPO-34 zeolites. Importantly, the dynamic process of reaction and diffusion process at the scale of individual zeolite crystal during MTO reaction can be clearly demonstrated. The inadequate usage of active sites and species, and the rapid deactivation,

which are two crucial puzzles for industrial MTO processes, can be directly interpreted. Substantially, the deep data approach can take the advantages of both the well-established theoretical model and specifically developed experimental techniques, and make it especially suitable for extensive applications where obtaining a complete picture of the process is not possible with only the simulations or experiments alone. In summary, we expect that this article may drive the attention of integration of visualization methods with simulations via the deep data approach to understand the evolution of molecules and active sites in heterogeneous catalysts.

## Methods
**Multiscale reaction–diffusion simulations**. The fixed-bed reactor filled with SAPO-34 zeolites is formulated at two scales, i.e., catalyst crystal and catalyst ensemble scale. At the catalyst crystal scale, the dynamics of MTO reaction is interpreted as the contributions of adsorption, diffusion and reaction kinetics[4,57], detailed mathematic modelling are introduced in Supplementary Equations 1–10. The effect of retained carbonaceous species in SAPO-34 on intracrystalline diffusivity of methane, methanol, ethylene, and propylene were investigated by MD. The details of MD simulations are introduced in Supplementary Figs. 14–16. The adsorption isotherms for gas components in SAPO-34 zeolites at low temperature were reported in our previous work[42,57]. The quantitative relation between adsorption isotherm and carbonaceous species deposited in SAPO-34 zeolites was measured by Intelligent Gravimetric Analyzer[57]. The intracrystalline diffusivities and diffusion activation energies were decoupled the effect of surface barriers[42]. The parameters at catalyst ensemble scale, e.g., catalyst lifetime, gas product distribution, relative quantity of retained acid sites and quantity of carbonaceous species, were used as feedback to determine the kinetic constants, the parameters are listed in Supplementary Table 3.

**Catalyst synthesis and physicochemical characterizations**. The synthesis procedures of SAPO-34 zeolites with different crystal size but similar Brønsted acidity and SAPO-34 zeolite with high silica content were taken from existing recipes from open literature[58,59]. The phase structure of the SAPO-34 was characterized by X-ray diffraction (XRD) (Supplementary Fig. 1). The distribution of crystal size and morphology were observed from field emission scanning electron microscope (FESEM) (Supplementary Table 1 and Supplementary Fig. 2). The statistical results of chemical composition of individual SAPO-34 zeolite crystals was measured by energy dispersive X-Ray spectroscopy (EDX) (Supplementary Table 1). The bulk

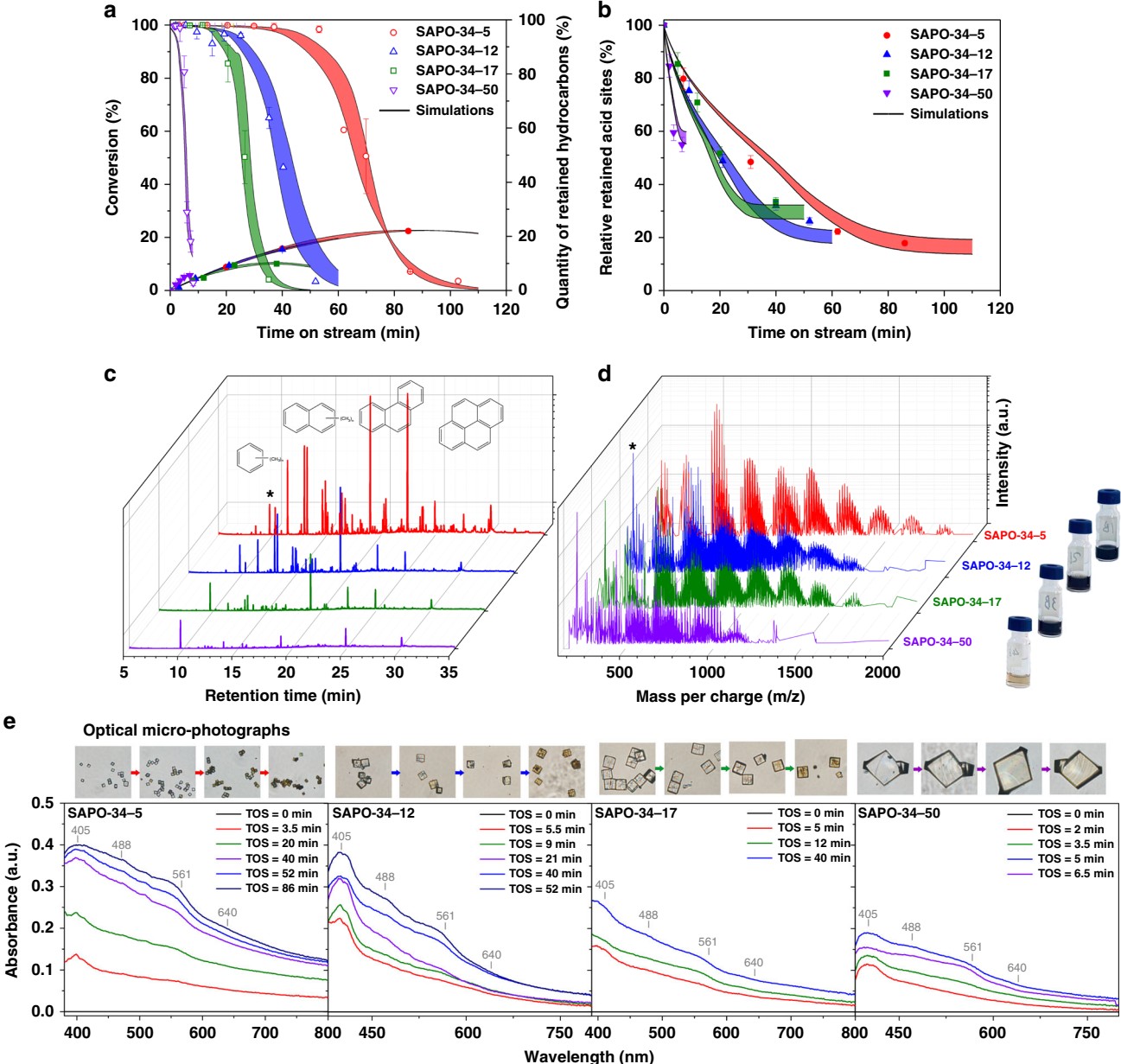

**Fig. 6 Multiple spectroscopy used for monitoring MTO reaction catalyzed by SAPO-34 zeolites. a** Methanol conversion analyzed by online GC and the content of retained carbonaceous species measured by TGA. **b** The relatively quantitative evolution of residual Brønsted acidity inside SAPO-34 zeolites measured by DRIFT spectra. The error band is the standard error of simulation results. **c** The component analysis of retained carbonaceous species, inside SAPO-34 zeolites after catalytic deactivation by GC-MS. * represents the internal standard. **d** The MALDI FT-ICR mass spectra of retained carbonaceous species with molecular weight larger than 200 Da. And the optical photographs of extracted phase of retained carbonaceous species in zeolites. **e** Optical micro-photographs and corresponding DR UV–vis spectra to reflect the discoloration of zeolite crystals and evolution of retained carbonaceous species, respectively.

chemical composition of SAPO-34 zeolite samples was analyzed by X-ray fluorescence (XRF) (Supplementary Table 1). The quantity of medium and strong acid sites of bulk SAPO-34 zeolite samples was measured by $NH_3$ temperature programmed desorption ($NH_3$-TPD) (Supplementary Table 1 and Supplementary Fig. 3). The $N_2$ adsorption/desorption was used to measure the textural property of SAPO-34 zeolites (Supplementary Fig. 4 and Supplementary Table 2).

**Catalytic performance of MTO reaction.** Catalytic testing of MTO was done in a fixed-bed quartz reactor with inner diameter 0.004 m at 723.2 ± 0.4 K. Weight-hourly space velocity (*WHSV*) of methanol was set to 5.0 ± 0.1 $g_{MeOH} \cdot g_{zeo.}^{-1} \cdot h^{-1}$ and partial pressure of methanol to 0.28 bar by flowing $N_2$. In order to ensure the relative uniform distribution of carbonaceous species along the catalyst bed (Supplementary Fig. 17), the high *WHSV* was used. Online analysis of the gas products was performed with an Agilent 7890B gas chromatography (GC)

equipped with FID detector and a PoraPLOT Q-HT capillary column. Detailed experimental results are shown in Supplementary Fig. 5.

**Characterization of acidity and carbonaceous species.** The time-evolution manners of average Brønsted acidity of bulk SAPO-34 zeolites during MTO reactions were recorded by diffuse reflectance infrared Fourier transform (DRIFT) spectra as shown in Supplementary Fig. 8.

The removal of retained carbonaceous species was measured by thermogravimetric analysis (TGA) and differential thermogravimetry (DTG) (DTG profiles are shown in Supplementary Fig. 9). DR (diffuse reflectance) UV/vis spectra[22,46] were performed with a VARIAN Cary-5000 UV–Vis-NIR spectrophotometer. Spent SAPO-34 zeolites were placed in PIKE cell with a temperature controller and its lid was equipped with quartz window. 15 mg of catalyst was dissolved in 1 mL of a 20 wt% HF solution in a Teflon container for 6 h[60]. The organic compounds were extracted by addition of 1 mL $CH_2Cl_2$ with

100 ppm internal standard $C_2Cl_6$ for 1 h. Analysis of the extracted phase was performed on an Agilent 7890A/5975C GC/MS instrument, equipped with a HP-5 capillary column and FID detector. Then the carbonaceous species in the extracted phase were then mixed with matrix 1,8,9-anthracenetriol and further analyzed by a 15-T SolariX XR FT-ICR MS (Bruker Daltonics)[55]. The instrument was equipped with a Nd:YAG laser ($\lambda = 335$ nm) and a time-of-flight mass analyzer in reflection mode. Positive ion mass spectra were recorded in the mass region between 200 and 3000 Da. The color changes of SAPO-34 zeolites used for MTO reaction were observed with an Olympus IX73 upright microscopy by using a ×40 0.6 NA high working-distance microscopy objective lens.

**Time-dependent density functional theory calculations**. In order to differentiate the carbonaceous species by SIM technique, TDDFT calculations[61] were performed to identify the excitation (first excitation energy, group state $S_0$) and emission (excited state $S_1$) wavelengths and understand the phosphorescence behavior of different carbonaceous species. Details are included in "time-dependent density functional theory" of Supplementary Information.

**Imaging of carbonaceous species in zeolite crystals by SIM**. The super-resolution imaging was carried out using a Nikon N-SIM super-resolution microscopy system with a motorized inverted microscope ECLIPSE Ti2-E, a ×100/NA 1.49 oil immersion TIRF objective lens (CFI HP) and ORCA-Flash 4.0 sCMOS camera (Hamamatsu Photonics K.K.)[20,21]. The wavelengths of illumination and emission detection of SIM used in this work are 405 (detection at 435–485 nm), 488 (detection at 500–545 nm), 561 (detection at 570–640 nm), and 640 nm (detection at 663–738 nm), respectively, which can cover the characteristic area of excitation and emission wavelengths of $B_n^+$, $N_n^+$, $PH_n^+$, and $PYR_n^+$. In the measurements, each illustration channel of SIM works independently, and the corresponding detector collects the light signal of emission. Images were taken at a Z-plane of middle of zeolitic crystal. The software NIS-Elements Ar and N-SIM Analysis were used to analyze the collected images and computationally reconstruct the super-resolution image as shown in Supplementary Figs. 10–13. To ensure the high-resolution of images can be obtained by SIM technique, the glass-bottomed culture dish (35-mm dish with 20-mm well) loaded with the sample has to be placed close to the objective lens. Therefore, in this work, the imaging experiments by SIM technique were not performed at reactive conditions to protect objective lens. The results of UV–vis spectra operated under reactive and non-reactive MTO conditions can potentially provide reference for SIM. Borodina et al.[47] found that the UV–vis spectra of carbonaceous species formed at high reaction temperatures (i.e., 573–773 K) show only minor changes after the cooling down to room temperature. Essentially the band of UV–vis spectra at 334 nm, which is assigned to low methylated benzene carbocations, shifts around 10 to 343 nm. Based on this, the imaging experimental results under non-reactive conditions might be used to reflect the situations under reactive conditions with good confidence, as evidenced by Fig. 3 and Supplementary Fig. 6, in which the simulated results agree well with SIM results in qualitative.

## Data availability
All data presented in this paper are available from the corresponding authors upon request.

## Code availability
The code used in this paper is available from the corresponding authors upon request.

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

## Acknowledgements

This work is supported by the National Natural Science Foundation of China (Grant No. 91834302) and Innovation program of science and research from Dalian Institute of Chemical Physics (DICP I201938). The authors thank Dr. Lihong Wan, Dr. Pengfei Wu, and Dr. Qinglong Qiao in Dalian Institute of Chemical Physics, Chinese Academy of Sciences for MALDI FT-ICR mass spectra characterization, synthesis of SAPO-34 zeolites, and discussion on fluorescence imaging.

## Author contributions

M.G. carried out the multiscale reaction–diffusion simulations, zeolitic material characterization, catalytic testing, and fluorescence imaging; H.L. and M.G. developed the procedure of reaction–diffusion simulation; M.G., W.L., Z.X., and Mao Y. visualized the retained carbonaceous species in individual zeolite crystals by SIM; S.P. helped to analyze the results of UV–vis and SIM; Miao Y. synthesized the SAPO-34 zeolites; M.G., H.L., and Mao Y. planed the experiments and simulations; Mao Y. and Z.L. initialized and supervised the project. All authors helped in writing the paper and commented on it.

## Competing interests

The authors declare no competing interests.
