## [Peer Review File · Nature Communications]

Reviewers' comments:

Reviewer #1 (Remarks to the Author):

The manuscript by Ye and Liu et al presents an alleged deep data approach to spatiotemporal understanding of key species in MTO chemistry using molecular simulations and (among a plethora of techniques) SIM microscopy. The main focus is on understanding differences between different sized SAPO particles. The topic and the aim is timely, and indeed, some mysteries are really worth exploring in MTO and its deactivation. The data is of high quality and seemingly all techniques (MD, characterization) are performed in a correct manner. The paper is well written and with a quite a good intro for the non-expert reader. It could be shortened a little, but length on average is justified.

There are however some major concerns (1, 2, 3):

1. I do question the main claim though, i.e. the authors claim a synergy of multiscale reaction diffusion simulation and microscopy (SIM) to give insight into the spatiotemporal evolution of key MTO species, and say this is a deep data approach.

The paper presents the word Deep data a lot on page 4, and in Fig 1, this is also somewhat visualized. However, no details on the deep data approach are found. Fig 1 shows some buzz words such as 'deep mechanisms...' and 'deep development' but all of this remains very vague. Later in the paper, they state: 'SIM images are used to verify the multi scale reaction diffusion simulation'. Is this the deep data? What is really deep? Is it just a comparison of the visual models with the SIM and then adjust some parameters? Is it with machine learning based iteration or prediction? ... This is not clear.

If I follow the thought that data of physical characterization is fed into the models, I have some questions (comment 2)

2. In Fig S2, clearly samples B and C have no real homogeneous particles sizes for the crystals. Also for the big 50 μ m one in D, only one crystal is shown. How is the size distribution of these crystals (synthesis) and how does it affect the authors' work where bulk physical characterization data (of a range of crystals with avg size e.g. + or - 10 μ m) is used as input in multi scale spatiotemporal evolution models of acidity, species and diffusion, for one crystal (e.g. in the presented visualizations (e.g. Fig 3, 4, 6b and 6d).

Similar questions for the DRIFTS, this is measured on bulk spent catalyst, but then used in single crystal computation (or not? This is not clear). Also, the figures give kg/m³ for acid sites: how come this is so high? (e.g. a zeolite with really a lot of acid sites (Si), has 1 mmol /g. Imagine this is a proton, weighing 1 g/mol, we get 1 mg/g zeolite. Then assuming a zeolite density of 1 g/mL (not correct but OK), we get 1 mg/mL zeolite. If we translate to m³, we get 1000 g/m³. But the scale bars in Fig. 3e for instance give around 150 kg/m³ at TOS 3; (so 150 times more than what a really acidic zeolite would have?). The authors should this rationalize their numbers a bit more and check them.

So, in general, how can one transfer bulk obtained information on materials to quantitative input for MD or other computational models centering on one crystal?

The link between All of this relates to the warning given for the deep data approach as explained in the benchmark outlook in Nature Materials (ref 23 in this work), that says "Despite simplicity, such an approach requires extreme care and expertise in interpreting the experimental data, to correct for instrumental artefacts (for example, calibration, giving systematic errors in measured structural parameters), partial character of information (for example, missing atoms of light elements poorly visible in electron microscopy), and systematic errors in theoretical models. Practically, this requires close interactions between theorists and experimentalists who are well aware of the strengths and limitations of their respective methods."

Can the authors briefly comment on how they made sure that they took 'extreme care' in experimental data given the partial character of information or systematic errors in theoretical models. These things should be addressed and the deep-data approach should be explained better (if it is really justified to claim it as such), i.e. more detailed and yet more understandable for the average reader. So, in general, how can one transfer bulk obtained information on materials to quantitative input for MD or other computational models centering on one crystal?

3. How does this open an interesting method for materials synthesis as claimed in the introduction? To claim this, and to show real promise of their approach (deep data or not), I suggest the authors to make one more SAPO-34 with a critical property change from synthesis (e.g. islanding Si, Si/Al+P variation, zoning, or something else) and test whether the deep data approach presented can indeed quickly offer insights that correlate with the catalytic data.

Minor comment 4: Along a similar line as comment 2, in terms of statistical analysis, little is found, e.g. were standard errors taken into consideration when measuring physical properties and is this error margin translated when the inputs were used for the MD models? Also some replications of key measurements used as input are needed with error estimation.

Minor comment 5: (Suppl table 1, two numbers after the dot for surface areas are not meaningful), Suppl fig 5, error bars on conversion? Same for Fig. 5 a and b in main manuscript.

Minor comment 6: fig 6 is in part discussed before fig 5 and this is confusing.

Finally, I wish to complement the authors on an impressive piece of work that is certainly valuable and with good insights (e.g. Fig 6, even if still a bit hypothetical, is very informative!). Given substantiation by the authors and demonstrating what exactly is the synergy (which data) and how the deep data approach is really used, this paper is of very high quality and very suited for Nature Communications, and a nice addition to the MTO catalysis field.

Reviewer #2 (Remarks to the Author):

Manuscript brings rather new insight into the properties and behaviour of SAPO-34 catalyst in the MTO process. It is an ambitious combination of simulations and space-resolved spectroscopy that together offers an interpretation of differences in catalytic activity of crystals with different size (ranging from 5 to 50 μm). Therefore, I believe it is appropriate for publication in Nature Communication. However, before it is published it should clarify and improve few details described below:

Authors talk about "deep data", however, I haven't found what exactly is meant by deep data. There are details about adsorption, diffusion and reactivity simulations, there are details about spectroscopy but I am missing the details how those two are put together. Please clarify.

Please provide more details about how kinetic constants summarized in SI Table 2 where obtained.

Are they fitted to provide agreement with experimental data reported in SI Fig. 5?

I have an objection on how the Figures are presented (2-4 and 6): concentration profile is shown as 2D plot showing the central plane of the whole crystal. However, profiles are symmetrical (and that is the natural consequence of the model underneath) and the same information can be presented as a simple function $y(x)$ where x is the distance from the edge to the centre of the crystal. Such figures do not need colour scheme and they (i) will bring clearer pictures, (ii) can combine more information in one figure and, thus, (iii) significantly reduce the space required without compromising clarity.

Figure 5, on the other hand, is too small and it is difficult to read, please organize it differently and make sure that reader can read axis and legends.

The level of English is visibly lower in SI compared to main text.

Supplementary Figures 15 – I do not see a need for 2D plots when the values are constant along x and colour scheme is needed (and extremely small values with unnecessary number of decimal points are used). Please change it to simple plot where y axis shows the value.
SI Figures 16-21 – same as above for Figures 2-4 and 6.

Reviewer #3 (Remarks to the Author):

The work by Gao et al. entitled Imaging spatiotemporal evolution of molecules and active sites in zeolite catalyst during methanol-to-olefins reaction contains the results from a fluorescence (SIM) microscopy study and modelling of diffusion behaviour in an attempt to understand how product and reactant diffusion affects the acid site availability and deactivation of catalytically relevant small pore zeolites for hydrocarbon conversion. The key premise in this work is that with techniques such as SIM it is now possible to study zeolite particles that are more relevant in size (industrially) when compared to what has been performed in the past and to subsequently be able to identify the species and their location that lead to 'real' deactivation. The results have been contextualised with more conventional lab-based characterisation methods and catalytic (time on stream testing). Indeed this sort of approach is likely to be of real interest to catalyst and even materials science researchers as they seek to obtain a better understanding of spatial-temporal effects in functional materials.

I am not really able to comment authoritatively on the diffusion modelling approach used but I do feel that there are two issues with the interpretation of the SIM data which need clarification. Firstly it is not clear to me how the illumination and detection strategy used here can really differentiate between the proposed chemical species. Whilst the premise is logical, i.e. that illumination with particular incident light will mainly excite molecules with the equivalent band gap (i.e. the authors say that methylbenzenes can be probed using an excitation wavelength of 405 nm), in effect there are many possible energy transfer mechanisms that means that molecules, particularly those that can be excited at a longer wavelength (> 405 nm), can also give a signal. Furthermore the images are constructed from a 'fluorescence' signal, but it has recently been shown that hydrocarbon species present in zeolites exhibit both fluorescence and phosphorescence (see <https://doi.org/10.1021/acs.jpcc.9b09050>). So I am wondering whether the authors can really differentiate/assign the chemical species based on their response to a (tunable) incident wavelength? Looking at the SIM figures, the distribution of species looks remarkably similar for the 488, 561 and 640 nm which suggests that actually it is not easy to differentiate between species as has been proposed.

A second issue concerns the lack of correlation with the UV-Vis data. All of the UV-Vis data for the crystals towards the end of the respective TOS studies contain very little absorbance at ~ 640 nm, yet according to the SIM fluorescence signal, this component represents the most prevalent species. Either then the imaging studies are not representative or else (again) there is an issue with the link with speciation. Or is there a problem with (fast) quenching of some species excited at shorter wavelengths? Notwithstanding that the extinction coefficient (even identity!) of the species present are not really known it is not a given that the intensity of an absorption band (ergo, an emission band) is an indication of the number of species present. As such it is not so easy to correlate the presence of particular species with deactivation phenomena. To be fair to the authors however, the simulations and citations of past work has been used intelligently to mitigate these risks but it is difficult to draw meaningful detailed conclusions from a rather straightforward analysis of fluorescence microscopy data. Also, if these experiments were not performed in a reactive atmosphere, it is known that the signals can become quenched in air – for example, emissivity begins to resemble that of graphene oxide in place of graphene.

Additional comments include; the EDX mapping of the crystals only indicates the elemental distribution at the crystal surface but not below the surface. As such this mapping is not proof of an uneven distribution of Si:Al.

Minor comments: Figure 1 is too general for the paper. For example, the authors discuss/highlight the possibilities of using XRD-CT (amongst other methods) but this isn't a review article so it doesn't need to be mentioned. Better to focus on what is being presented in the article only.

Response to reviewers' comments

Reviewers' comments:

We would like to thank the editor and reviewers for their constructive comments. We have carefully examined each comment and made necessary changes to address the concerns following the reviewers' suggestion and improve our manuscript.

Reviewer #1 (Remarks to the Author):

The manuscript by Ye and Liu et al presents an alleged deep data approach to spatiotemporal understanding of key species in MTO chemistry using molecular simulations and (among a plethora of techniques) SIM microscopy. The main focus is on understanding differences between different sized SAPO particles. The topic and the aim is timely, and indeed, some mysteries are really worth exploring in MTO and its deactivation. The data is of high quality and seemingly all techniques (MD, characterization) are performed in a correct manner. The paper is well written and with a quite a good intro for the non-expert reader. It could be shortened a little, but length on average is justified.

Response: We thank the reviewer for the positive comments. We have elaborated the key issues concerning the methods used in this work and improved our manuscript according to the reviewers' suggestions. We have modified the manuscript to make

the contents more concise, especially the presentations of concepts and figures more clearly.

There are however some major concerns (1, 2, 3):

1. I do question the main claim though, i.e. the authors claim a synergy of multiscale reaction diffusion simulation and microscopy (SIM) to give insight into the spatiotemporal evolution of key MTO species, and say this is a deep data approach. The paper presents the word Deep data a lot on page 4, and in Fig 1, this is also somewhat visualized. However, no details on the deep data approach are found. Fig 1 shows some buzz words such as ‘deep mechanisms...’ and ‘deep development’ but all of this remains very vague. Later in the paper, they state: ‘SIM images are used to verify the multi scale reaction diffusion simulation’. Is this the deep data? What is really deep? Is it just a comparison of the visual models with the SIM and then adjust some parameters? Is it with machine learning based iteration or prediction? ... This is not clear.

Response 1: We thank the reviewer for the comment. In this work we used the term deep data following Kalinin et al.¹. Here deep data refers to an approach aiming to provide physical or chemical insights behind experimental data and fills in the missing information that is hard to obtain via available experimental techniques¹. It is different from the big data analytics frequently used in computer science, which mainly utilizes the unsupervised learning or machine learning to derive the

correlations or search for optimal solutions based on a large collection of data. The idea underlying deep data is to merge elaborative experimental analysis into an established theoretical model, first to estimate the input parameters and validate the model, and then to provide further predictions on the detailed evolution and/or dynamics of a specific process at the length-scale concerned. Owing to the practical importance in light olefins production, methanol-to-olefins (MTO) reaction has attracted a wide variety of researchers to understand the underlying reaction mechanism at molecular scale and establish the relationship between catalyst properties and reaction performance at bulk material scale. However, despite that detailed knowledge at these two separate scales have been richly accumulated, catalyst lifetime and product distribution, for example, are seldomly related to the evolution of carbonaceous species and catalyst acidity in a direct way due to the absence of advanced in-situ measurement techniques linking these two scales. In the deep data approach considered in this work (as shown in Figure 1a), a multiscale reaction-diffusion model, developed at single catalyst crystal scale and taken into consideration of mass transfer between catalysts and gas phase at catalyst ensemble scale, provides a link of MTO reaction observed at different length-scales. In the model, the change of molecular loading is described by reaction kinetics and molecular flux via the reaction-diffusion equation (Supplementary Equation 1). The reaction kinetics is developed based on the dual-cycle mechanism². The molecular flux is calculated by Maxwell-Stefan equation (Supplementary Equation 2) and ideal adsorbed solution theory, which needs the input of molecular diffusivities, Langmuir

adsorption parameters and adsorption enthalpy of the zeolitic framework. In addition, the crystal size and quantity of acid sites are two fundamental parameters required to solve the reaction-diffusion equation for individual catalyst crystal. The molecular flux at the catalyst surface is then used to connect the mass transfer between crystal and gas phase. Given that the properties of individual catalyst crystals and molecular flux at the crystal surface are known, the catalyst ensemble will be simulated by well-established fixed bed model (Supplementary Equation 10), which the effects of space velocity and partial pressure of methanol can be considered. Thus it is possible, by modelling, to comprehend the reaction performance at catalyst ensemble scale on the basis of structures and properties of host materials, evolution of guest molecules, and interplay between host materials and guest molecules at catalyst crystal scale. Normally it is extremely difficult to measure, e.g. the spatiotemporal distribution of acid sites, and evolution of gas molecules inside individual catalyst crystal, though these parameters are vital in gaining the insights into MTO reactions. The multiscale reaction-diffusion model in the deep data approach, on the meantime, is quite promising in simulating the dynamic process of the evolution of acid sites and gas molecules inside individual catalyst crystal in MTO reaction. In this sense, the deep data approach is expected to shed lights on the mechanism of MTO reactions over zeolite catalysts supposed that some key parameters at catalyst crystal scale such as guest molecular diffusivities, adsorption isotherms, crystal size, quantity of acid sites and rate constants of reaction kinetics can be carefully obtained via molecular dynamics simulations, as well as various measurements. Note that most of

experiments are conducted with catalyst ensemble, the measured parameters need to be transferred from over bulk materials to that over individual SAPO-34 zeolite crystals, which will be introduced in “**Response 4**” in details. The parameters of reaction kinetics, as it is hard to directly obtain at single zeolite scale, were obtained by fitting modelling results with experimental data at catalyst ensemble scale, in terms of catalyst lifetime, product distributions and evolution of acid sites and carbonaceous species on bulk SAPO-34. The parameters are then used as input for the multiscale reaction-diffusion model in simulating MTO reaction over individual catalyst crystal, which is validated with the advent of structural illumination microscopy (SIM) with the regard to the spatiotemporal evolution of carbonaceous species inside SAPO-34 zeolite crystal. With the validated multiscale reaction-diffusion modelling, the deep data approach can fill in the missing information of MTO reactions at concerned length-scale, e.g. spatiotemporal evolution of acid sites, gas molecules inside catalyst crystal, and provide insights to MTO reaction catalyzed by SAPO-34 zeolites (see “**Response 6**”).

We have supplemented the discussion above in Page 4 in the “Introduction” and Page 7 in the section “Implement of deep data approach to MTO reactions” of manuscript based on Figure 1a.

Figure 1a. Implementation of the deep data approach to MTO reactions: merging experimental data into multiscale reaction-diffusion modelling. M is measurable, C is calculable and F is the feedback from experimental data at catalyst ensemble.

If I follow the thought that data of physical characterization is fed into the models, I have some questions (comment 2)

2. In Fig S2, clearly samples B and C have no real homogeneous particles sizes for the crystals. Also, for the big 50 μm one in D, only one crystal is shown. How is the size distribution of these crystals (synthesis) and how does it affect the authors' work where bulk physical characterization data (of a range of crystals with avg size e.g. + or - 10 μm) is used as input in multi scale spatiotemporal evolution models of acidity, species and diffusion, for one crystal (e.g. in the presented visualizations (e.g. Fig 3, 4, 6b and 6d).

Response 2: We would like to thank the reviewer for raising the query. Indeed, the crystal size of SAPO-34 zeolites is one of the key parameters for MTO reactions and

also affects the simulation results. To this end, we controlled the crystal size of SAPO-34 zeolite samples by regulating synthetic formulation before experiments³. The crystal size distributions of SAPO-34 zeolite crystals were monitored by field emission scanning electron microscopy (FE-SEM), as shown in Supplementary Figure 2. The number of crystals used for statistics of crystal size distribution is larger than 80 ~ 120 for each measurement. We found that the crystal size is $4.82 \pm 0.36 \mu\text{m}$ for SAPO-34-5, $11.17 \pm 1.80 \mu\text{m}$ for SAPO-34-12, $11.68 \pm 1.22 \mu\text{m}$ for SAPO-34-12_HighSi, $16.92 \pm 1.66 \mu\text{m}$ for SAPO-34-17 and $47.08 \pm 3.50 \mu\text{m}$ for SAPO-34-50. The maximum deviations of crystal size for all SAPO-34 zeolite samples were controlled within 15%. The upper and lower limit of crystal size of SAPO-34 zeolite samples were then separately used as input parameters of the model to examine the effect of size distribution on the simulated results. As shown in Figures 3 and Figure 6a-b, it manifests that the size distribution of SAPO-34 crystal has only a minor influence on the simulation results due to relatively uniform crystal size of the sample used.

We have added the crystal size distribution of SAPO-34 zeolite samples in Supplementary Table 1 and Supplementary Figure 2. We also added the standard errors of simulation results caused by size distribution in Figure 3, Figure 4 and Figure 6a-b, Supplementary Figure 5 and Supplementary Figure 17-19.

Supplementary Figure 2. Representative FE-SEM images of **a** SAPO-34-5, **c** SAPO-34-12, **e** SAPO-34-17, **g** SAPO-34-50 and **i** SAPO-34-12_HighSi samples. Crystal size distribution of **b** SAPO-34-5, **d** SAPO-34-12, **f** SAPO-34-17, **h** SAPO-34-50 and **j** SAPO-34-12_HighSi samples based on the statistics over 80-120 individual crystals.

Figure 3. The spatiotemporal distribution of carbonaceous species obtained from SIM and simulations in **a, c** SAPO-34-5 ($4.82 \pm 0.36 \mu\text{m}$), **b, d** SAPO-34-12 ($11.17 \pm 1.80 \mu\text{m}$), **e, g** SAPO-34-17 ($16.92 \pm 1.66 \mu\text{m}$), **f, h** SAPO-34-50 ($47.08 \pm 3.50 \mu\text{m}$) samples during MTO reactions at 723 K and $WHSV$ of $5.0 \pm 0.1 \text{ g}_{\text{MeOH}} \cdot \text{g}_{\text{Zco}}^{-1} \cdot \text{h}^{-1}$. The quantity of acid sites is $1.00 \pm 0.06 \text{ mmol/g}_{\text{Zco}}$. The error band is standard error of simulation results. The SIM images shown is the fluorescence that originated from the overlap of four profiles with a laser excitation of 405 nm (detection at 435-485 nm, false color: blue), 488 nm (detection at 500-545 nm, false color: green), 561 nm (detection at 570-640 nm, false color: red), 640 nm (detection at 663-738 nm, false color: pink). The images were taken in the middle plane of the zeolite crystal. The scale bar represents $10 \mu\text{m}$.

Figure 6a-b. The effect of distribution of crystal size and quantity of acid sites on the MTO reaction catalyzed by **a** SAPO-34-5 ($4.82 \pm 0.36 \mu\text{m}$), **b** SAPO-34-12 ($11.17 \pm 1.80 \mu\text{m}$), **e** SAPO-34-17 ($16.92 \pm 1.66 \mu\text{m}$) and **f** SAPO-34-50 ($47.08 \pm 3.50 \mu\text{m}$) samples at 723 K by simulations. The quantity of acid sites is $1.00 \pm 0.06 \text{ mmol/g}_{\text{zeo}}$. The $WHSV$ is $5.0 \pm 0.1 \text{ g}_{\text{MeOH}} \cdot \text{g}_{\text{zeo}}^{-1} \cdot \text{h}^{-1}$. The error band is standard error of simulation results.

Similar questions for the DRIFTS, this is measured on bulk spent catalyst, but then used in single crystal computation (or not? This is not clear). Also, the figures give kg/m^3 for acid sites: how come this is so high? (e.g. a zeolite with really a lot of acid sites (Si), has 1 mmol/g . Imagine this is a proton, weighing 1 g/mol , we get 1 mg/g zeolite. Then assuming a zeolite density of 1 g/mL (not correct but OK), we get 1 mg/mL zeolite. If we translate to m^3 , we get 1000 g/m^3 . But the scale bars in Fig. 3e for instance give around 150 kg/m^3 at TOS = 3; (so 150 times more than what a really acidic zeolite would have?). The authors should this rationalize their numbers a bit more and check them.

Response 3: We thank the reviewer for careful reading. The diffuse reflectance infrared Fourier transform (DRIFT) was employed to measure the changes in average acidity of bulk SAPO-34 zeolites during MTO reactions. The measured results of

acidity by DRIFT were used as input for reaction-diffusion modelling at the catalyst ensemble scale as shown in Figure 1a. The spatiotemporal evolution of acid sites inside single SAPO-34 zeolite crystal was predicted by the simulations.

As to the difference of acid sites between the estimations by the reviewer and simulation results, it comes from the different molecular weight used for representing the single acid site. In our kinetic model, for simplicity, we defined a virtual HCP species that is a lump of active carbonaceous species covering the acid sites. In this way, we assume that 1 mol HCP species would cover 1 mol acid site, and use a virtual molecular weight of acid site of 140 g/mol, i.e. the average molecular weight of HCP species, in the simulations. We agree with the reviewer that physically the molecular weight of a single acid site might be close to that of a proton, i.e. 1.0 g/mol. The virtual molecular weight, despite being easily implemented in the model, is overestimated. But in the simulations, the acid sites do not appear in the mass balance of hydrocarbon conversions, and the virtual molecular weight of acid sites will not affect the simulation results of the quantity of acid sites for an individual SAPO-34 zeolite crystal. In our simulations, we obtained that the quantity of acid sites for an individual SAPO-34 zeolite crystal is about 1.00 ± 0.06 mmol/g_{zeo.}, which agrees well with the estimation by the reviewer.

Note that it may cause misleading to utilize the unit of kg/m³ to account for the quantity of acid sites as in our original manuscript, we have modified the unit of the quantity of acid sites from kg/m³ to mmol/g_{zeo.} in Figure 4e, and added the explanations on the virtual molecular weight of acid sites in Page 11 in the section

“Experiments and reaction-diffusion simulations of MTO reaction” of Supplementary Information. In addition, we have supplemented the illustration of the average of bulk SAPO-34 zeolites measured by DRIFT and NH₃ temperature programmed desorption (NH₃-TPD) in Page 26 in the section ‘Method’ of manuscript.

So, in general, how can one transfer bulk obtained information on materials to quantitative input for MD or other computational models centering on one crystal?

Response 4: Thank the reviewer for the question. As shown in Figure 1a, the main parameters at catalyst crystal scale include crystal size, quantity of acid sites, molecular diffusivities and adsorption isotherms.

The crystal size of SAPO-34 zeolite samples was controlled via regulating synthetic formulation before experiments. The distribution crystal size measured by FE-SEM was shown to be relatively uniform based on the statistics over 80-120 individual SAPO-34 zeolite crystals. We obtained that the crystal size is $4.82 \pm 0.36 \mu\text{m}$ for SAPO-34-5, $11.17 \pm 1.80 \mu\text{m}$ for SAPO-34-12, $11.68 \pm 1.22 \mu\text{m}$ for SAPO-34-12_HighSi, $16.92 \pm 1.66 \mu\text{m}$ for SAPO-34-17 and $47.08 \pm 3.50 \mu\text{m}$ for SAPO-34-50. The maximum deviations of crystal size for all SAPO-34 zeolite samples were controlled within 15%. Thus, the average crystal size is used as input for the simulations and the effect of distribution of crystal size was also subsequently examined. Please also refer to the “**Response 2**”.

We measured the average quantity of acid sites and chemical composition of bulk

SAPO-34 zeolite samples by NH_3 -temperature programmed desorption (NH_3 -TPD) and X-Ray fluorescence (XRF), respectively, which are shown in Supplementary Table 1. Based on the results we obtained about per 0.1 silica content corresponds 1 $\text{mmol/g}_{\text{zeo}}$ acid sites. We then used energy dispersive X-Ray fluorescence spectrometer (EDX) to measure the chemical composition (silica content) of individual SAPO-34 zeolite crystals. It shows relatively narrow distribution of silica content as in Supplementary Table 1, and we obtained that the silica contents are 0.096 ± 0.001 for SAPO-34-5, 0.091 ± 0.003 for SAPO-34-12, 0.170 ± 0.005 for SAPO-34-12_HighSi, 0.103 ± 0.003 for SAPO-34-17, and 0.110 ± 0.006 for SAPO-34-50. The maximum deviations of silica content of individual SAPO-34 zeolite crystals for all samples are 6%. Further comparison of silica contents measured by XRF and EDX reflects the similar composition between crystal surface and bulk crystals. By use of relation between silica content and quantity of acid sites, i.e. per 0.1 silica content corresponding to 1 $\text{mmol/g}_{\text{zeo}}$ acid sites, the quantity of acid sites for individual crystal was estimated. The influences of distribution of quantity of acid sites on simulation results were also subsequently examined.

The molecular diffusivities and adsorption isotherms were obtained from either molecular dynamics simulations or measured by uptake experiments over bulk SAPO-34 zeolite samples with narrow distribution of crystal size. As discussed in our previous work⁴, for a narrow distribution of crystal size, the molecular diffusivities over bulk SAPO-34 zeolite samples can be used for individual zeolite crystal, which is verified by pulsed field gradient (PFG) NMR measurements. The adsorption

isotherms are thermodynamic parameters that we used the adsorption parameters and heat over bulk samples as input for individual crystals.

We have supplemented the method that transferring material properties obtained on bulk SAPO-34 zeolites to that in individual zeolite crystal in details in Page 8 in the section of “Implement of deep data approach to MTO reactions” of manuscript, Page 2-6 and Page 30 of Supplementary Information. We have added the results of crystal size, quantity of acid sites and chemical composition in the Supplementary Table 1.

Supplementary Table 1. Statistical results of crystal size and silica content of individual SAPO-34 zeolite crystals and average quantity of acid sites and silica content of bulk SAPO-34 zeolite samples.

Samples	Crystal size (μm)	Silica content by EDX	Silica content by XRF	Bulk acidity by $\text{NH}_3\text{-TPD}$ ($\text{mmol/g}_{\text{zeo.}}$)
SAPO-34-5	4.82 ± 0.36	$\text{Si}_{0.096 \pm 0.001}$	$\text{Si}_{0.097}$	1.01 ± 0.01
SAPO-34-12	11.17 ± 1.80	$\text{Si}_{0.091 \pm 0.003}$	$\text{Si}_{0.093}$	1.09 ± 0.02
SAPO-34-12_HighSi	11.68 ± 1.22	$\text{Si}_{0.170 \pm 0.005}$	$\text{Si}_{0.170}$	1.72 ± 0.02
SAPO-34-17	16.92 ± 1.66	$\text{Si}_{0.103 \pm 0.003}$	$\text{Si}_{0.092}$	1.05 ± 0.01
SAPO-34-50	47.08 ± 3.50	$\text{Si}_{0.110 \pm 0.006}$	$\text{Si}_{0.105}$	1.02 ± 0.02

The link between All of this relates to the warning given for the deep data approach as explained in the benchmark outlook in *Nature Materials* (ref 23 in this work), that says “Despite simplicity, such an approach requires extreme care and expertise in

interpreting the experimental data, to correct for instrumental artefacts (for example, calibration, giving systematic errors in measured structural parameters), partial character of information (for example, missing atoms of light elements poorly visible in electron microscopy), and systematic errors in theoretical models. Practically, this requires close interactions between theorists and experimentalists who are well aware of the strengths and limitations of their respective methods.”

Can the authors briefly comment on how they made sure that they took ‘extreme care’ in experimental data given the partial character of information or systematic errors in theoretical models. These things should be addressed and the deep-data approach should be explained better (if it is really justified to claim it as such), i.e. more detailed and yet more understandable for the average reader. So, in general, how can one transfer bulk obtained information on materials to quantitative input for MD or other computational models centering on one crystal?

Response 5: Thank the reviewer for valuable suggestion. We fully agree with that the deep data approach requires extreme care in deriving the experimental data and analyzing the systematic errors in the theoretical model. As shown in Figure 1a, especially the input parameters and corresponding errors at the catalyst crystal scale need to be examined carefully.

In doing so, first of all, it is very important to prepare SAPO-34 zeolite crystal sample with a relatively uniform distribution of crystal, as it is crucial for the simulations and measurements. In this work, the maximum deviations of crystal size

for all SAPO-34 zeolite samples were controlled within 15%, and the systematic errors of simulations caused by crystal size distribution were evaluated as shown in Supplementary Table 4.

The second important parameter that was carefully controlled is the silica contents of SAPO-34 zeolite crystals. The silica content of individual SAPO-34 zeolite crystals for all samples were measured by EDX and the maximum deviations of 6% were achieved, as shown in Supplementary Table 1. Further to that, the silica contents of bulk SAPO-34 zeolite samples were also monitored by XRF, which are in accordance with that measured by EDX as in Supplementary Table 1, reflecting the consistent silica contents of individual crystal and bulk samples. Note that the silica content is directly related to the quantity of acid sites, we argue that the distribution of acid sites of individual SAPO-34 zeolite crystal is relatively uniform, and the quantity of acid sites between surface and bulk crystals is approximately the same. The systematic errors of simulations caused by acid sites distribution were evaluated and shown in Supplementary Table 4.

In addition, the CHA topology and micropore structure of all SAPO-34 zeolite samples were tested and shown in Supplementary Figure 1 and Supplementary Figure 4, respectively, which are premise for molecular dynamics simulations. For MTO reactions, the errors of *WHSV* and reaction temperature were controlled within 2% and 0.2%, and the systematic errors of simulations caused by *WHSV* were evaluated and shown in Supplementary Table 4.

In summary, as shown in Supplementary Table 4, in this work, the systematic errors

of simulations caused by material properties and reaction conditions are carefully examined. In general, to transfer material properties at catalyst ensemble scale to that at catalyst crystal scale, relatively uniform distributions of crystal size and quantity of acid sites of individual zeolite crystals are the key. Well-defined the CHA topology and crystalline structure are essential for molecular dynamics simulations, particularly with regard to the derivation of molecular diffusivities and adsorption. In addition, reaction conditions over catalyst ensemble need to be carefully controlled as well.

Supplementary Table 4. Average standard error of simulations of methanol conversion, selectivity of ethylene, propylene, C₄₊ and alkanes, relative quantity of retained acidity and coke content caused by error of crystal size, acidity and *WHSV*.

Error margin	Conversion	Sel. C ₂ ⁼	Sel. C ₃ ⁼	Sel. C ₄ ⁼	Sel. Alk.	Acidity	Coke
± 10% crystal size	2.82	0.19	0.27	0.27	0.39	1.22	0.23
± 15% crystal size	5.32	0.87	1.57	0.64	2.60	1.90	0.36
± 10% acidity	0.99	0.13	0.15	0.20	0.27	1.59	0.32
± 10% WHSV	3.13	0.14	0.21	0.20	0.33	1.12	0.22

We have supplemented the standard errors of the material properties at catalyst crystal scale, as well as systematic errors of MTO reaction results over catalyst ensemble, in Page 9 in the section “Implement of deep data approach to MTO reactions” of manuscript. The discussion on the transferring measurement results of bulk material to that of individual catalyst crystal are also added in the same section of manuscript. The evaluations of systematic errors of simulations caused by these

parameters are also added in the Supplementary Table 4.

3. How does this open an interesting method for materials synthesis as claimed in the introduction? To claim this, and to show real promise of their approach (deep data or not), I suggest the authors to make one more SAPO-34 with a critical property change from synthesis (e.g. islanding Si, Si/Al+P variation, zoning, or something else) and test whether the deep data approach presented can indeed quickly offer insights that correlate with the catalytic data.

Response 6: Thank the reviewer for helpful suggestions. We have synthesized a new SAPO-34 zeolite sample with high silica content to test the influence of silica content on MTO reaction. The new sample, namely SAPO-34-12_HighSi, has average crystal size of $11.68 \pm 1.22 \mu\text{m}$, chemical composition of $\text{Al}_{0.426 \pm 0.019} \text{P}_{0.401 \pm 0.011} \text{Si}_{0.170 \pm 0.005}$ and quantity of acid sites of $1.72 \pm 0.02 \text{ mmol/g}_{\text{zeo}}$. Relatively uniform distributions of crystal size and silica content of SAPO-34-12_HighSi are also granted (see Supplementary Table 1). Based on the estimated parameters, the MTO reaction catalyzed by SAPO-34-12_HighSi was simulated. The simulated results, as shown in Supplementary Figure 6, manifest that increasing silica content of SAPO-34 zeolites, i.e. quantity of acid sites, will shorten catalyst lifetime, decrease initial selectivity of ethylene, and increase initial selectivity of C_{4+} , which agree well with the experimental results. Compared with results of SAPO-34-12, higher relative quantity of acid sites (59.54%) and less quantity of carbonaceous species (16.75 wt%) retained

in deactivated SAPO-34-12_HighSi sample can be observed. To understand MTO reaction catalyzed by SAPO-34-12_High Si, the simulated spatiotemporal evolution of molecules and acid sites during MTO reaction are shown in Supplementary Figure 6e and Supplementary Figure 7. In particular, the spatiotemporal evolution of HCP species and coke precursors inside SAPO-34-12_HighSi zeolite crystal during MTO reaction has been verified by SIM results as shown in Supplementary Figure 6d. As can be seen in Supplementary Figure 6a and e, more quantity of alkanes, HCP species and coke precursors are formed at the initial stage of MTO reaction over SAPO-34-12_HighSi than that over SAPO-34-12. This can be explained as that a higher quantity of acid sites would promote the occurrence of cyclization and hydrogen transfer reaction. We also observed from Supplementary Figure 6e that, at the initial stage, increasing the quantity of acid sites causes HCP species and coke precursors are accumulated at the region close to rim ($\sim 2 \mu\text{m}$ from the rim). Part of olefinic products generated at this region can diffuse toward the crystal center and be rapidly converted to HCP species and then coke precursors. Therefore, the accumulation of coke precursors at the crystal center can be observed. Subsequently, at the crystal rim, high concentrated HCP species can rapidly react with methanol and olefinic products to further form coke precursors, and coke precursors formed at the rim significantly hinder the diffusion of reactant into crystal interior. In this connection, in the deactivated SAPO-34-12_HighSi sample, plenty of acid sites and HCP species trapped in the crystal are inaccessible for reactants, which can be found in Supplementary Figure 6b and c. By use of the multiscale reaction-diffusion

simulations, the insights of insufficient utilization of acid sites and HCP species during MTO reaction catalyzed by SAPO-34 with high silica content are unveiled.

We have supplemented above simulated and experimental results in Page 13 in the section “Spatiotemporal evolution of carbonaceous species in SAPO-34” of manuscript and Page 14-18 in the section “Experiments and reaction-diffusion simulations of MTO reaction” of Supplementary Information as an extended study case to illustrate that the deep data approach can be potentially used to predict the influence of catalyst properties, which can be altered via material synthesis, on MTO reaction.

Supplementary Figure 6. Comparison of MTO reactions catalyzed by SAPO-34-12_HighSi sample ($11.68 \pm 1.22 \mu\text{m}$) between simulated and experimental results. **a** Conversion of methanol and selectivity of gas products. **b** Evolution of quantity of retained acid sites and carbonaceous species in bulk SAPO-34 zeolites. **c** Component analysis of retained carbonaceous species inside SAPO-34-12 and SAPO-34-12_HighSi zeolite samples after catalytic deactivation by GC-MS. * represents the internal standard. **d** Spatiotemporal evolution of carbonaceous species inside crystal obtained by SIM. **e** Simulated results of spatiotemporal evolution of HCP species and coke precursors inside crystal. The quantity of acid sites is $1.70 \pm 0.10 \text{ mmol/g}_{\text{zeo}}$. The error band is standard error of simulation results. Experimental conditions: $T = 723 \text{ K}$, $WHSV = 5.0 \pm 0.1 \text{ g}_{\text{MeOH}} \cdot \text{g}_{\text{zeo}}^{-1} \cdot \text{h}^{-1}$, partial pressure of methanol of 0.28 bar.

Supplementary Figure 7. Simulated results of loading of **a** methanol, **b** propylene, **c** C_{4+} and **d** acid sites and reaction rate of **e** HCP species and **f** coke precursors inside SAPO-34 zeolite crystal during MTO reactions catalyzed by SAPO-34-12_HighSi sample. The quantity of acid sites is 1.70 ± 0.10 mmol/g_{zeo.}. $WHSV$ is 5.0 ± 0.1 g_{MeOH}·g_{zeo.}⁻¹·h⁻¹. The error band is standard error of simulation results.

Minor comment 4: Along a similar line as comment 2, in terms of statistical analysis, little is found, e.g. were standard errors taken into consideration when measuring physical properties and is this error margin translated when the inputs were used for the MD models? Also some replications of key measurements used as input are needed with error estimation.

Response 7: We thank the reviewer for the suggestions. The statistical analysis of crystal size and silica content of individual SAPO-34 zeolite crystals were measured. According to the distribution of silica content of individual crystal (within 6% deviation) and relation between silica content and quantity of acid sites, the maximum

error margin of 6% is taken for the quantity of acid sites, i.e. 1.00 ± 0.06 mmol/g_{zeo.}. Additionally, *WHSV* of MTO reactions is 5.0 ± 0.1 g_{MeOH}·g_{zeo.}⁻¹·h⁻¹. The impact of standard errors of these parameters on simulations have been given in Figure 3, Figure 4, Figure 6a-b, Supplementary Figure 5, Supplementary Figure 18 and Supplementary Figure 19.

We have supplemented the effects of error margin of crystal size, quantity of acid sites of SAPO-34 zeolites and *WHSV* of MTO reactions on the simulations.

Minor comment 5: (Suppl table 1, two numbers after the dot for surface areas are not meaningful), Suppl fig 5, error bars on conversion? Same for Fig. 5 a and b in main manuscript.

Response 8: We thank the reviewer for the suggestions. We have modified the significant figures of surface area in Supplementary Table 2. We have added the standard errors of simulated and experimental methanol conversion, product selectivity, coke content and acidity in Supplementary Figure 5 and Figure 6a and b.

Minor comment 6: fig 6 is in part discussed before fig 5 and this is confusing.

Response 9: We thank the reviewer for the suggestion. We have adjusted the order of discussions of Figure 6 (now is Figure 5) before that on Figure 5 (now is Figure 6) to avoid confusing the readers.

Finally, I wish to complement the authors on an impressive piece of work that is certainly valuable and with good insights (e.g. Fig 6, even if still a bit hypothetical, is very informative!). Given substantiation by the authors and demonstrating what exactly is the synergy (which data) and how the deep data approach is really used, this paper is of very high quality and very suited for *Nature Communications*, and a nice addition to the MTO catalysis field.

Response 10: We thank the reviewer again for recognizing the value of this work. We have revised the manuscript according to the suggestions of reviewer, we believe that the quality of manuscript has been further improved.

Reviewer #2 (Remarks to the Author):

Manuscript brings rather new insight into the properties and behaviour of SAPO-34 catalyst in the MTO process. It is an ambitious combination of simulations and space-resolved spectroscopy that together offers an interpretation of differences in catalytic activity of crystals with different size (ranging from 5 to 50 μm). Therefore, I believe it is appropriate for publication in *Nature Communications*. However, before it is published it should clarify and improve few details described below:

Response: We are appreciative of the reviewer's recognition and recommendation to this work.

Authors talk about "deep data", however, I haven't found what exactly is meant by deep data. There are details about adsorption, diffusion and reactivity simulations, there are details about spectroscopy but I am missing the details how those two are put together. Please clarify.

Response 1: We thank the reviewer for careful reading. The idea underlying deep data is to merge elaborative experimental analysis into an established theoretical model, first to estimate the input parameters and validate the model, and then to provide further predictions on the detailed evolution and/or dynamics of a specific process at the length-scale concerned. In the deep data approach considered in this

work (as shown in Figure 1a), a multiscale reaction-diffusion model, developed at single catalyst crystal scale and taken into consideration of mass transfer between catalysts and gas phase at ensemble scale, provides a link of MTO reaction observed at different length scales. Normally it is extremely difficult to measure, e.g. the spatiotemporal distribution of acid sites, and evolution of gas molecules inside individual catalyst crystal, though these parameters are vital in gaining the insights into MTO reactions. The multiscale reaction-diffusion model in the deep data approach, on the meantime, is quite promising in simulating the dynamic process of the evolution of acid sites and gas molecules inside individual catalyst crystal in MTO reaction. In the model, the change of molecular loading is described by reaction kinetics and molecular flux via the reaction-diffusion equation (Supplementary Equation 1). The reaction kinetics is developed based on the dual-cycle mechanism². The molecular flux is calculated by Maxwell-Stefan equation (Supplementary Equation 2) and ideal absorbed solution theory, which needs the input of molecular diffusivities, Langmuir adsorption parameters and adsorption enthalpy of the zeolitic framework. In addition, the crystal size and quantity of acid sites are two fundamental parameters required to solve the reaction-diffusion equation for individual catalyst crystal. The molecular flux at the catalyst surface is then used to connect the mass transfer between crystal and gas phase. Given that the properties of individual catalyst crystals and molecular flux at the crystal surface are known, the catalyst ensemble will be simulated by well-established fixed bed model (Supplementary Equation 10), which the effects of space velocity and partial pressure of methanol can be considered.

In this sense, the deep data approach is expected to shed lights on the mechanism of MTO reactions over zeolite catalysts supposed that some key parameters at catalyst crystal scale such as guest molecular diffusivities and adsorption isotherms, crystal size, quantity of acid sites and reaction kinetics can be carefully obtained via molecular dynamics simulations, as well as various measurements. Therefore, it requires extreme care in deriving the experimental data and analyzing the systematic errors in the model¹. First of all, preparing SAPO-34 zeolite crystal sample with a relatively uniform distribution of crystal size and acid sites is crucial for the simulations and measurements. The CHA topologies and micropore structures of SAPO-34 samples are also premise for molecular dynamics (MD) simulations, which were carefully tested and shown in Supplementary Figure 1 and Supplementary Figure 4. The molecular diffusivities and adsorption isotherms were obtained either from MD or uptake experiments⁴. For all SAPO-34 zeolite samples, the maximum deviation of 15% was achieved for crystal size and 6% for silica content, as shown in Supplementary Table 1. Note that ~ 0.1 silica content corresponds 1 mmol/g_{zeo.} acid sites (Supplementary Table 1), the quantity of acid sites for individual crystal of samples was estimated to be 1.00 ± 0.06 mmol/g_{zeo.}. The parameters of reaction kinetics, as it is hard to directly obtain at single zeolite scale, were obtained by fitting modelling results with experimental data at catalyst ensemble scale, in terms of catalyst lifetime, product distributions (catalytic testing) and evolution of acid sites (DRIFT) and carbonaceous species (TGA combined with dissolution experiments in quantitative and DR UV-vis in semiquantitative) on bulk SAPO-34. The parameters

are then used as input for the multiscale reaction-diffusion model in simulating MTO reaction over individual catalyst crystal, which is validated with the advent of structural illumination microscopy (SIM) with regard to the spatiotemporal evolution of carbonaceous species inside SAPO-34 zeolite crystal. With the validated multiscale reaction-diffusion modelling, the deep data approach can fill in the missing information of MTO reactions at concerned length-scale and provide insights to MTO reaction catalyzed by SAPO-34 zeolites.

We have supplemented the description of deep data approach and illustrated the method in Page 7 in the section “Implement of deep data approach to MTO reactions” of manuscript.

Figure 1a. Implementation of the deep data approach to MTO reactions: merging experimental data into multiscale reaction-diffusion modelling. M is measurable, C is calculable and F is the feedback from experimental data at catalyst ensemble.

Please provide more details about how kinetic constants summarized in SI Table 2 where obtained. Are they fitted to provide agreement with experimental data reported in SI Fig. 5?

Response 2: We would like to thank the reviewer for raising the query. The reaction kinetic constants listed in Supplementary Table 2 (now is Supplementary Table 3) were obtained by fitting experimental data with SAPO-34-5, SAPO-34-12, SAPO-34-17 and SAPO-34-50 samples, respectively, as shown in Supplementary Figure 5 and Fig.6 a and b.

Note that it is hard, if not impossible, to directly verify the kinetic constants, we have examined the rationality of kinetic constants against the results of density functional theory (DFT) calculations^{5, 6, 7}. DFT calculations show that the propylene is more likely to be formed than ethylene during olefins-cycle^{5, 6}, and we indeed found that the reaction kinetic constant of propylene formation is larger than that of ethylene formation in the olefins-cycle. We obtained that the kinetic constant of ethylene formation is larger than that of propylene formation in aromatics-cycle, which are well consistent with the DFT calculations⁷ that show that the overall free energy of ethylene formation is lower than that of propylene formation. MTO reaction catalyzed by SAPO-34-12_HighSi sample (high silica content) as an extended study case further validates the applicability of kinetic constants as shown in Supplementary Figure 6.

We have supplemented the details of estimation of kinetic constants listed in

Supplementary Table 3 in Page 9 in the section of “Implement of deep data approach to MTO reactions” of manuscript and in Page 12 in the section of “Experiments and reaction-diffusion simulations of MTO reaction” of Supplementary Information.

I have an objection on how the Figures are presented (2-4 and 6): concentration profile is shown as 2D plot showing the central plane of the whole crystal. However, profiles are symmetrical (and that is the natural consequence of the model underneath) and the same information can be presented as a simple function $y(x)$ where x is the distance from the edge to the centre of the crystal. Such figures do not need colour scheme and they (i) will bring clearer pictures, (ii) can combine more information in one figure and, thus, (iii) significantly reduce the space required without compromising clarity.

Response 3: We agree with the reviewer’s suggestions. We have modified the 2D plots (Figure 3-5 of manuscript) to figures presented as a function $y(x)$, with x being the distance from the edge to the centre of the crystal. The 2D plots of X-Y plane of center section of crystals in Figure 5b and c, however, are still retained, in order to summarize the effect of crystal size on the evolutions of acid sites, carbonaceous species and molecular diffusion during MTO reactions.

Figure 3. The spatiotemporal distribution of carbonaceous species obtained from SIM and simulations in **a, c** SAPO-34-5 ($4.82 \pm 0.36 \mu\text{m}$), **b, d** SAPO-34-12 ($11.17 \pm 1.80 \mu\text{m}$), **e, g** SAPO-34-17 ($16.92 \pm 1.66 \mu\text{m}$), **f, h** SAPO-34-50 ($47.08 \pm 3.50 \mu\text{m}$) samples during MTO reactions at 723 K and $WHSV$ of $5.0 \pm 0.1 \text{ g}_{\text{MeOH}} \cdot \text{g}_{\text{Zco.}}^{-1} \cdot \text{h}^{-1}$. The quantity of acid sites is $1.00 \pm 0.06 \text{ mmol/g}_{\text{Zco.}}$. The error band is standard error of simulation results. The SIM images shown is the fluorescence that originated from the overlap of four profiles with a laser excitation of 405 nm (detection at 435-485 nm, false color: blue), 488 nm (detection at 500-545 nm, false color: green), 561 nm (detection at 570-640 nm, false color: red), 640 nm (detection at 663-738 nm, false color: pink). The images were taken in the middle plane of the zeolite crystal. The scale bar represents $10 \mu\text{m}$.

Figure 4. The spatiotemporal evolution of concentration of **a** methanol, **b** ethylene, **c** propylene, **d** C_{4+} and **e** acid sites in SAPO-34-5, SAPO-34-12, SAPO-34-17 and SAPO-34-50 crystals during MTO reactions at 723 K and $WHSV$ of $5.0 \pm 0.1 \text{ g}_{\text{MeOH}} \cdot \text{g}_{\text{ZCO}}^{-1} \cdot \text{h}^{-1}$. The quantity of acid sites is $1.00 \pm 0.06 \text{ mmol/g}_{\text{ZCO}}$. The results obtained from multiscale reaction-diffusion simulations. The error band is standard error of simulation results.

Figure 5. Molecular reaction-diffusion mechanism during MTO reaction on the individual zeolite crystal level. **a** The evolution of coke precursors at the rim of SAPO-34 zeolites with different crystal size. **b** and **c** Schematics of spatiotemporal evolution of acid sites, HCP species and coke precursors during MTO reaction in small and large crystal size, respectively. 2D plots represent a quarter of X-Y plane of center section of crystal. The curves present the loading of acid sites and carbonaceous species as the function of the distance from the centre to edge of the crystal.

Figure 5, on the other hand, is too small and it is difficult to read, please organize it differently and make sure that reader can read axis and legends.

Response 4: We appreciate the suggestion. We have rearranged the Figure 6 and improve the resolution to make the axis and legends readable.

Figure 6. The results of multiple spectroscopic techniques used for monitoring MTO reaction catalyzed by SAPO-34 zeolites with different crystal size. **a**, methanol conversion analyzed by online GC and the content of retained carbonaceous species measured by TGA. **b**, the quantitative evolution of residual Brønsted acidity inside SAPO-34 zeolites measured by DRIFT spectra. The lines represent the simulated results by reaction-diffusion simulations. The error band is standard error of simulations. **c**, the analysis of retained carbonaceous species, i.e. methylbenzenes, methylnaphthalene, phenanthrene and pyrene, inside SAPO-34 zeolites after catalytic deactivation by GC-MS. * represents the internal standard. **d**, the MALDI FT-ICR mass spectra of retained carbonaceous species with molecular weight larger than 200 Da. * represents the internal standard. And optical photographs of extracted phase of retained carbonaceous species in zeolites. **e**, optical micro-photographs and corresponding DR UV-vis spectra to reflect the discoloration of zeolite crystals and evolution of retained carbonaceous species, respectively.

The level of English is visibly lower in SI compared to main text.

Response 5: We have improved the language in the Supplementary Information.

Supplementary Figures 15 – I do not see a need for 2D plots when the values are constant along x and colour scheme is needed (and extremely small values with unnecessary number of decimal points are used). Please change it to simple plot where y axis shows the value.

Response 6: We have modified the 2D plots in the Supplementary Figure 17.

Supplementary Figure 17. Concentration profile of retained carbonaceous species along the axial direction of reactor during MTO reaction catalyzed by SAPO-34-12 zeolite sample. Simulation conditions: $T = 723$ K, $WHSV = 5.0 \pm 0.1$ $\text{g}_{\text{MeOH}} \cdot \text{g}_{\text{zeo.}}^{-1} \cdot \text{h}^{-1}$, partial pressure of methanol of 0.28 bar.

SI Figures 16-21 – same as above for Figures 2-4 and 6.

Response: We have modified the 2D plots in the Supplementary Figure 18 and 19.

Supplementary Figure 18. The simulated spatiotemporal evolution of reaction rate of **a** methanol, **b** HCP species and **c** coke precursors in SAPO-34-5, SAPO-34-12, SAPO-34-17 and SAPO-34-50 crystals during MTO reactions at 723 K and $WHSV$ of $5.0 \pm 0.1 \text{ g}_{\text{MeOH}} \cdot \text{g}_{\text{ZCO}}^{-1} \cdot \text{h}^{-1}$. The quantity of acid sites is $1.00 \pm 0.06 \text{ mmol/g}_{\text{ZCO}}$. The error band is standard error of simulation results. The sign of reaction rate of methanol is negative, which represents the consumption of methanol.

Supplementary Figure 19. The simulated spatiotemporal evolution of diffusion flux of **a** ethylene, **b** propylene, **c** C_{4+} in SAPO-34-5, SAPO-34-12, SAPO-34-17 and SAPO-34-50 crystals during MTO reactions at 723 K and $WHSV$ of $5.0 \pm 0.1 \text{ g}_{MeOH} \cdot \text{g}_{zeo}^{-1} \cdot \text{h}^{-1}$. The quantity of acid sites is $1.00 \pm 0.06 \text{ mmol/g}_{zeo}$. The error band is standard error of simulation results. The positive sign of diffusion flux represents chemicals diffusion toward crystal center, and negative sign of diffusion flux represents chemicals diffusion outward the crystal.

Reviewer #3 (Remarks to the Author):

The work by Gao et al. entitled Imaging spatiotemporal evolution of molecules and active sites in zeolite catalyst during methanol-to-olefins reaction contains the results from a fluorescence (SIM) microscopy study and modelling of diffusion behaviour in an attempt to understand how product and reactant diffusion affects the acid site availability and deactivation of catalytically relevant small pore zeolites for hydrocarbon conversion. The key premise in this work is that with techniques such as SIM it is now possible to study zeolite particles that are more relevant in size (industrially) when compared to what has been performed in the past and to subsequently be able to identify the species and their location that lead to 'real' deactivation. The results have been contextualised with more conventional lab-based characterisation methods and catalytic (time on stream testing). Indeed this sort of approach is likely to be of real interest to catalyst and even materials science researchers as they seek to obtain a better understanding of spatial-temporal effects in functional materials.

Response: We appreciate the reviewer for the recognition.

I am not really able to comment authoritatively on the diffusion modelling approach used but I do feel that there are two issues with the interpretation of the SIM data which need clarification. Firstly it is not clear to me how the illumination and

detection strategy used here can really differentiate between the proposed chemical species. Whilst the premise is logical, i.e. that illumination with particular incident light will mainly excite molecules with the equivalent band gap (i.e. the authors say that methylbenzenes can be probed using an excitation wavelength of 405 nm), in effect there are many possible energy transfer mechanisms that means that molecules, particularly those that can be excited at a longer wavelength (> 405 nm), can also give a signal. Furthermore the images are constructed from a 'fluorescence' signal, but it has recently been shown that hydrocarbon species present in zeolites exhibit both fluorescence and phosphorescence (see <https://doi.org/10.1021/acs.jpcc.9b09050>). So I am wondering whether the authors can really differentiate/assign the chemical species based on their response to a (tunable) incident wavelength? Looking at the SIM figures, the distribution of species looks remarkably similar for the 488, 561 and 640 nm which suggests that actually it is not easy to differentiate between species as has been proposed.

Response 1: We thank the reviewer for the careful reading. In order to differentiate the carbonaceous species by structural illumination imaging (SIM) technique, we first performed time-dependent density functional theory (TDDFT) calculations to identify the excitation (first excitation energy, ground state S_0) and emission (excited state S_1) wavelengths of different carbonaceous species. In doing so, all the structure optimizations were carried out without constraints at B3LYP/6-31G (d, p) levels, for both the ground and excited states of carbonaceous species in gas phase⁸. The

harmonic frequency calculations based on Gaussian 09 package⁹ confirmed that the optimized structures correspond to the minimum-energy points with all the frequencies being positive. In the calculations, the confinement imposed by CHA topology was not considered owing to the very time-consuming computation. It is found that the deviation of excitation wavelength at confined state from that of gas phase carbonaceous species is about 30 nm⁸. The summaries of the calculated excitation and emission wavelengths of benzenic (B_n^+), naphthalenic (N_n^+), phenanthrenic (PH_n^+) and pyrenic (PYR_n^+) carbocations with n methyl substituents are shown in Figure 2 and Supplementary Table 5. As can be seen, the excitation wavelengths of B_n^+ , N_n^+ , PH_n^+ and PYR_n^+ are situated around 390, 480, 560 and 640 nm, respectively, and the corresponding emission wavelengths are located in the range of 480-490, 500-520, 620-630 and 670-700 nm, respectively. The wavelengths of illumination and emission detection of SIM used in this work are 405 (detection at 435-485 nm), 488 (detection at 500-545 nm), 561 (detection at 570-640 nm) and 640 nm (detection at 663-738 nm), respectively, which can cover the characteristic area of excitation and emission wavelengths of B_n^+ , N_n^+ , PH_n^+ and PYR_n^+ . In the measurements, each illumination channel of SIM works independently and the corresponding detector collects the light signal of emission. In MTO reactions over SAPO-34 zeolites, however, methylbenzenes B_n^+ and methylnaphthalene N_n^+ are shown to be activated carbonaceous species (i.e. HCP species)¹⁰, while phenanthrene PH_n^+ and pyrene PYR_n^+ are considered to be coke precursors¹⁰. As can be seen in Figure 2, therefore, it is feasible to differentiate the HCP species (B_n^+ and N_n^+) and

coke precursors (PH_n^+ and PYR_n^+) inside SAPO-34 zeolites by SIM. However, we agree that the identification of each single carbonaceous species via SIM, at current stage, is still quite challenging.

Additionally, we performed TDDFT to estimate the phosphorescence behavior of carbonaceous species (state T_1). The phosphorescence signals of B_2^+ , N_4^+ , PH_0^+ and PYR_0^+ show a peak at 578, 568, 700 and 816 nm, respectively. Compared to that of fluorescence signal, the wavelength of phosphorescence signal for given species is about 80 nm higher. It has been similarly observed by time resolved photoluminescence spectroscopy of the trimethyladamantylammonium hydroxide confined in **CHA** topology¹¹. In this work, we argue that since the wavelengths of illumination and detection of SIM are sufficiently close (i.e. the wavelengths of excitation at 405, 488, 561 and 640 nm and the wavelengths of detection at 435-485, 500-545, 570-640 and 663-738 nm, respectively), the interference of phosphorescence could be neglected. For example, as in Supplementary Table 5, if the phenanthrenic carbocations are excited at 561 nm, the fluorescence signal can be captured by the detector with working range of 570-640 nm while the phosphorescence signal with peak at ~ 700 nm can be hardly measured by the same detector.

To further validate that simulated results well accord with SIM images, we have added a study case of MTO reactions catalyzed by SAPO-34-12_HighSi in Supplementary Figure 6. We have supplemented the strategy of illumination and emission detection to differentiate between HCP species and coke precursors by SIM in Page 10 in the section “Spatiotemporal evolution of carbonaceous species in

SAPO-34”, Page 28 in the section “Methods” of manuscript and Page 24 in the section “Super resolution structured illumination microscopy” of Supplementary Information. In addition, the calculated results by TDDFT are added in Page 22 in the section “Time-dependent density functional theory” of Supplementary Information, Figure 2 and Supplementary Table 5. We have modified the statement that relation between excited wavelength and specific carbonaceous species in the caption of Figure 3 and Supplementary Information. The discussion about phosphorescence phenomenon has been added in Page 11 in the section “Spatiotemporal evolution of carbonaceous species in SAPO-34” of manuscript and Page 22 in the section “Super resolution structured illumination microscopy” of Supplementary Information and we have cited “Omori, N. et al. Understanding the Dynamics of Fluorescence Emission during Zeolite Detemplation Using Time Resolved Photoluminescence Spectroscopy. *J. Phys. Chem. C* **124**, 531-543 (2020)” as reviewer suggested in the in reference [50] of manuscript.

Figure 2. Simulated excitation (first excitation energies, group state S_0)^{8, 12}, emission (excited states S_1) wavelengths and corresponding oscillator strength of charged carbonaceous species in gas phase calculated at the B3LYP/6-31G (d, p) level of theory (see also Supplementary Table 5). The lines (or bands) around 405 (435-485 nm), 488 (500-545 nm), 561 (570-640 nm) and 640 nm

(663-738 nm) used as wavelength of excitation (emission detection) by SIM are indicated in blue, green, red and pink, respectively. B_n^+ , N_n^+ , PH_n^+ and PYR_n^+ stand for benzenic, naphthalenic, phenanthrenic and pyrenic carbocation with n methyl substituents, respectively.

Supplementary Table 5. Wavelengths of excitation (first excitation energies, group state S_0)^{8, 12} and emission (excited states S_1) of charged carbonaceous species in gas phase calculated at the B3LYP/6-31G (d, p) level of theory.

Species	Excitation wavelength (nm)	Excitation wavelength (nm) ^a	Oscillator strength of absorbance (-)	Emission wavelength (nm)	Oscillator strength of emission (-)
B_1^+	311	342	0.0799	482	0.0001
B_2^+	317	368	0.1058	484	0.0002
B_5^+	345	391	0.0751	492	0.0002
B_6^+	330	385	0.0778	499	0.0002
N_0^+	447	466	0.0024	503	0.0044
N_1^+	441	494	0.0003	496	0.0018
N_3^+	448	469	0.0027	503	0.0047
N_4^+	452	520	0.0002	508	0.0005
N_5^+	455	/	0.0014	515	0.0012
PH_0^+	550	559	0.0009	632	0.0021
PH_1^+	552	/	0.0009	623	0.0023
PYR_0^+	625	623	0.0005	672	0.0015
PYR_1^+	659	/	0.0007	665	0.0006

^a Calculated results of TDDFT^{8, 12}, which were considered the confinement by CHA topology.

A second issue concerns the lack of correlation with the UV-Vis data. All of the UV-Vis data for the crystals towards the end of the respective TOS studies contain very little absorbance at ~ 640 nm, yet according to the SIM fluorescence signal, this component represents the most prevalent species. Either then the imaging studies are not representative or else (again) there is an issue with the link with speciation. Or is there a problem with (fast) quenching of some species excited at shorter wavelengths? Notwithstanding that the extinction coefficient (even identity!) of the species present are not really known it is not a given that the intensity of an absorption band (ergo, an emission band) is an indication of the number of species present. As such it is not so easy to correlate the presence of particular species with deactivation phenomena. To be fair to the authors however, the simulations and citations of past work has been used intelligently to mitigate these risks but it is difficult to draw meaningful detailed conclusions from a rather straightforward analysis of fluorescence microscopy data. Also, if these experiments were not performed in a reactive atmosphere, it is known that the signals can become quenched in air – for example, emissivity begins to resemble that of graphene oxide in place of graphene.

Response 2: We thank the reviewer for the comments. We agree that it is difficult to directly correlate the results of UV-vis spectra with SIM images. Firstly, the absorbed and emissive responses of a given carbonaceous species might be different even if it is excited at the same wavelength. Figure 2a shows the oscillator strength that represents the absorbance of carbonaceous species to excited wavelength, which manifests an

overall decrease from B_n^+ , N_n^+ , PH_n^+ to PYR_n^+ . On the meantime, in Figure 2b, the oscillator strength reflects the luminescent intensity of excited carbonaceous species shows an overall increase from B_n^+ , N_n^+ , PH_n^+ to PYR_n^+ . These results indicate that the absorbance of a given carbonaceous species (e.g. PYR_n^+) to incident light at the wavelength of, for instance 640 nm, is weak while the emitted fluorescence signal excited by light at the same wavelength is much stronger. Therefore, the direct comparison between UV-vis spectra and SIM images is quite challenge. Secondly, in this work, SIM images and UV-vis spectra are used to verify the modelling results with different purposes. The fluorescence image obtained by SIM can qualitatively illustrate spatial distribution of a given type of carbonaceous species excited at a certain wavelength. There are four wavelengths, i.e. 405 (detection at 435-485 nm), 488 (detection at 500-545 nm), 561 (detection at 570-640 nm) and 640 nm (detection at 663-738 nm), used for B_n^+ , N_n^+ , PH_n^+ to PYR_n^+ , respectively. In this way, the spatial distribution of relative concentration of either HCP species or coke precursors within an SAPO-34 zeolite crystal at certain reaction stage can be obtained. However, due to the wavelengths of incident lights are different, the SIM images of one carbonaceous species cannot be directly compared to SIM images of another species. UV-vis spectra, on the other hand, provide semiquantitative comparison of a given type of carbonaceous species excited at the same wavelength, e.g. 405, 488, 561 or 640 nm, for different SAPO-34 zeolite samples at different MTO reaction stages. In this measurement, the temporal evolutions of HCP species and coke precursors inside SAPO-34 zeolite samples are mainly concerned. The quantities of carbonaceous

species were also obtained by thermogravimetric analysis (TGA) combined with dissolution/extraction experiments. Thirdly, the SIM and UV-vis spectra results are separately compared to the simulation work and partially validate the multi-scale reaction-diffusion model. The model is used to link the spatial and temporal evolution of carbonaceous species. As realized by the reviewer, this is hard to be achieved solely by the measurement techniques. In addition to the carbonaceous species, the spatial and temporal distributions of acid sites inside SAPO-34 zeolites can also be derived via modelling. This on the meantime reflects the effectiveness and significance of the deep data approach.

To ensure the high-resolution of images can be obtained by SIM technique, the glass-bottomed culture dish loaded with the sample is required to close to a $100\times/NA$ 1.49 oil immersion TIRF objective lens. Therefore, in this work, the imaging experiments by SIM technique were not performed at reactive conditions to protect objective lens. The results of UV-vis spectra operated under reactive and non-reactive MTO conditions can potentially provide reference for SIM. Borodina et al.¹⁰ found that the UV-vis spectra of carbonaceous species formed at high reaction temperatures (i.e. 573-773 K) show only a minor changes after the cooling down to room temperature. Essentially the band of UV-vis spectra at 334 nm, which is assigned to low methylated benzene carbocations, shifts only around 10 nm to 343 nm. Based on this, the imaging experimental results under non-reactive conditions might be used to reflect the situations under reactive conditions with good confidence, as evidenced by Figure 3 and Supplementary Figure 6, in which the simulated results agree well with

SIM results in qualitative. It should be stressed, however, SIM measurements under reactive conditions with a careful-designed experimental facility is highly desired for further study.

We have supplemented the discussions on the discrepant results between UV-vis spectra and SIM in Page 21 in the section “Interpretation of macroscopic phenomena by microscopic images” of manuscript and Page 25 in the section “Super resolution structured illumination microscopy” of Supplementary Information. We have added the reason and potential impacts of SIM performed under non-reactive conditions in Page 28 in the “Method” of manuscript. And we have emphasized the images obtained by SIM were used to qualitatively compare with multiscale reaction-diffusion simulations in Page 12 in the section “Spatiotemporal evolution of carbonaceous species in SAPO-34” of manuscript. We have modified the statement that comparison of fluorescence intensity between carbonaceous species excited at different wavelength in the manuscript.

Additional comments include; the EDX mapping of the crystals only indicates the elemental distribution at the crystal surface but not below the surface. As such this mapping is not proof of an uneven distribution of Si: Al.

Response 3: We thank the reviewer for good suggestion. In addition to EDX, we have supplemented the measurements of chemical composition on bulk material by X-ray fluorescence (XRF). As shown in Supplementary Table 1, the chemical compositions

on bulk and external surface are similar for all SAPO-34 zeolite samples. We have added silica content in the Supplementary Table 1 and detailed chemical composition in Page 3 in the section “Characterization of the SAPO-34 zeolites” of Supplementary Information.

Supplementary Table 1. Statistical results of silica content of individual SAPO-34 zeolite crystals and average silica content of bulk SAPO-34 zeolite samples.

Samples	Silica content by EDX	Silica content by XRF
SAPO-34-5	Si _{0.096±0.001}	Si _{0.097}
SAPO-34-12	Si _{0.091±0.003}	Si _{0.093}
SAPO-34-12_HighSi	Si _{0.170±0.005}	Si _{0.170}
SAPO-34-17	Si _{0.103±0.003}	Si _{0.092}
SAPO-34-50	Si _{0.110±0.006}	Si _{0.105}

Minor comments: Figure 1 is too general for the paper. For example, the authors discuss/highlight the possibilities of using XRD-CT (amongst other methods) but this isn't a review article so it doesn't need to be mentioned. Better to focus on what is being presented in the article only.

Response 4: We thank the reviewer for suggestion. We have modified the Figure 1a accordingly. The explanation can be also found in the “**Response 1**” to the reviewer 1.

Figure 1. Implementation of the deep data approach to MTO reactions: merging experimental data into multiscale reaction-diffusion modelling. M is measurable, C is calculable and F is the feedback from experimental data at catalyst ensemble.

Reference in response letter

1. Kalinin, S. V., Sumpter, B. G. & Archibald, R. K. Big-deep-smart data in imaging for guiding materials design. *Nat. Mater.* **14**, 973 (2015).
2. Hereijgers, B. P. C. *et al.* Product shape selectivity dominates the Methanol-to-Olefins (MTO) reaction over H-SAPO-34 catalysts. *J. Catal.* **264**, 77-87 (2009).
3. Yang, M. *et al.* A top-down approach to prepare silicoaluminophosphate molecular sieve nanocrystals with improved catalytic activity. *Chem. Commun.* **50**, 1845-1847 (2014).
4. Gao, M. *et al.* Direct quantification of surface barriers for mass transfer in nanoporous crystalline materials. *Commun. Chem.* **2**, 43 (2019).
5. Wang, C.-M., Wang, Y.-D. & Xie, Z.-K. Insights into the reaction mechanism of methanol-to-olefins conversion in HSAPO-34 from first principles: Are olefins themselves the dominating hydrocarbon pool species? *J. Catal.* **301**, 8-19 (2013).
6. Wang, S. *et al.* Polymethylbenzene or Alkene Cycle? Theoretical Study on Their Contribution to the Process of Methanol to Olefins over H-ZSM-5 Zeolite. *J. Phys. Chem. C* **119**, 28482-28498 (2015).
7. De Wispelaere, K., Hemelsoet, K., Waroquier, M. & Van Speybroeck, V. Complete low-barrier side-chain route for olefin formation during methanol conversion in H-SAPO-34. *J. Catal.* **305**, 76-80 (2013).
8. Hemelsoet, K. *et al.* Identification of Intermediates in Zeolite-Catalyzed Reactions by In Situ UV/Vis Microspectroscopy and a Complementary Set of Molecular Simulations. *Chem. Eur. J.* **19**, 16595-16606 (2013).
9. M. J. Frisch *et al.* *Gaussian 09*, Revision B.01 edn (Gaussian, Inc., Wallingford CT, 2016).
10. Borodina, E. *et al.* Influence of the Reaction Temperature on the Nature of the Active and Deactivating Species during Methanol to Olefins Conversion over H-SSZ-13. *ACS Catal.* **5**, 992-1003 (2015).
11. Omori, N. *et al.* Understanding the Dynamics of Fluorescence Emission during Zeolite Detemplation Using Time Resolved Photoluminescence Spectroscopy. *J. Phys. Chem. C* **124**, 531-543 (2020).
12. Van Speybroeck, V. *et al.* Mechanistic Studies on Chabazite-Type Methanol-to-Olefin Catalysts: Insights from Time-Resolved UV/Vis Microspectroscopy Combined with Theoretical Simulations. *ChemCatChem* **5**, 173-184 (2012).

REVIEWERS' COMMENTS:

Reviewer #1 (Remarks to the Author):

The authors did a thorough job of adapting their paper to the many comments, increasing its clearness and accuracy.

The first 2 responses to reviewer n° 3, I cannot check since they are not in my field of expertise (and the addition of Fig 2).

In some captions, they mention:

"The error band is standard error of simulation results."

this is explained in the letter to the reviewers, but may not be clear to the reader. Perhaps explicit where the error originates?

Apart from that the work is excellent and ready for publication in my opinion.

Reviewer #2 (Remarks to the Author):

Authors have very carefully answered all comments of all three reviewers and they have made major changes to both, main text and supporting information. I believe that all answers as well as modification of the text (and figures in particular) are sufficient and I can recommend the manuscript for publication in its present form. In particular I am satisfied with improved description of "deep data" approach and with the section on assignment of excitation and emission energies based on TDDFT calculations.

Reviewer #3 (Remarks to the Author):

I have been through the revised version of the article entitled 'Imaging spatiotemporal evolution of molecules and active sites in zeolite catalyst during methanol-to-olefins reaction' and would like to say firstly that the authors have clearly spent a lot of effort on updating the manuscript taking into account the feedback from the first round of reviewing and this has resulted in a more considered piece of work. I have though the following comments that require addressing before I think it can be published:

1) The TDDFT section is a welcome addition to the manuscript although the nomenclature used to identify the various species present in the sample is unclear to a non-expert like myself. For example in the supplementary information, Table 5 and according to the explanation proffered on page 26, I interpret a B2+ species to be a naphthenic species (i.e. 2 Benzene molecules). However there are NO+ species assigned so I am not sure what B2+ is supposed to signify. I presume (methyl?) substituted benzenes but this needs to be better articulated.

2) Although there is more welcome information on how the SIM data were accumulated it is still not clear what the time difference was from when the SIM measurements were performed and how long samples had been exposed to the atmosphere after being removed from the MTO reactor. As I mentioned in the first round of reviewing, we have seen that such species will react with O₂ and whatever else is in the air and that this changes their composition, ergo their emission response. And of course this will undermine the findings of this study (note in the study of Borodina et al. the implication according to the TOC/Figure 1 is that the samples are interrogated in situ i.e. from within the reactor and whilst there are changes as a function of temperature, there is no evidence that the

same changes occur on exposure to an oxidising atmosphere). However, I think it is unfair to ask the authors to solve this problem as it is very challenging and beyond the scope of the study and that the real value of these results is that it serves as a demonstrator case. Besides which there has been a lot of effort to cross correlate the findings from SIM with other data (modelling, UV-Vis). I do think though that some comment regarding the time taken to acquire the SIM data and the time the samples were exposed to the air should be included lest others who embark on similar studies will end up laboring under the false pretense that all that is required is a SIM setup + few reacted samples. My PhD student who has performed similar studies in the past would argue that unless the atmosphere was controlled, then what you are measuring are not the species that are present during the reaction. Fortunately she is not reviewing this manuscript.

Response to the reviewers' comments

Reviewers' comments:

We would like to thank again the editor and reviewers for their constructive comments. We have carefully examined each comment and made necessary changes to address the concerns following the reviewers' suggestion.

Reviewer #1 (Remarks to the Author):

The authors did a thorough job of adapting their paper to the many comments, increasing its clearness and accuracy.

The first 2 responses to reviewer n° 3, I cannot check since they are not in my field of expertise (and the addition of Fig 2).

In some captions, they mention: "The error band is standard error of simulation results." this is explained in the letter to the reviewers, but may not be clear to the reader. Perhaps explicit where the error originates?

Apart from that the work is excellent and ready for publication in my opinion.

Response: We appreciate the reviewer for his/her effort in reviewing our manuscript, as well as his/her supportive comments on this work. We have added "Fig. 3 also shows the standard error of simulations, which might be caused by the distribution of crystal size and acid sites (Supplementary Table 1) and/or error in estimating *WHSV*" in Page 13 in the revised manuscript to clarify the origin of errors.

Reviewer #2 (Remarks to the Author):

Authors have very carefully answered all comments of all three reviewers and they have made major changes to both, main text and supporting information. I believe that all answers as well as modification of the text (and figures in particular) are sufficient and I can recommend the manuscript for publication in its present form. In particular I am satisfied with improved description of "deep data" approach and with the section on assignment of excitation and emission energies based on TDDFT calculations.

Response: We appreciate the reviewer for his/her effort in reviewing our manuscript, as well as his/her positive comments on this work.

Reviewer #3 (Remarks to the Author):

I have been through the revised version of the article entitled ‘Imaging spatiotemporal evolution of molecules and active sites in zeolite catalyst during methanol-to-olefins reaction’ and would like to say firstly that the authors have clearly spent a lot of effort on updating the manuscript taking into account the feedback from the first round of reviewing and this has resulted in a more considered piece of work. I have though the following comments that require addressing before I think it can be published:

1) The TDDFT section is a welcome addition to the manuscript although the nomenclature used to identify the various species present in the sample is unclear to a non-expert like myself. For example, in the supplementary information, Table 5 and according to the explanation proffered on page 26, I interpret a B_2^+ species to be a naphthenic species (i.e. 2 Benzene molecules). However, there are N_0^+ species assigned so I am not sure what B_2^+ is supposed to signify. I presume (methyl?) substituted benzenes but this needs to be better articulated.

Response: We appreciate the reviewer for his/her effort in reviewing our manuscript, as well as his/her positive comments on this work. We have added statement of “ B_n^+ , N_n^+ , PH_n^+ and PYR_n^+ stand for benzenic, naphthalenic, phenanthrenic and pyrenic carbocation with n methyl substituents, respectively” in Supplementary Table 5 in the revised Supplementary Information to clarify that the subscript “ n ” represents the n methyl substituents.

2) Although there is more welcome information on how the SIM data were accumulated it is still not clear what the time difference was from when the SIM measurements were performed and how long samples had been exposed to the atmosphere after being removed from the MTO reactor. As I mentioned in the first round of reviewing, we have seen that such species will react with O_2 and whatever else is in the air and that this changes their composition, ergo their emission response. And of course this will undermine the findings of this study (note in the study of

Borodina et al. the implication according to the TOC/Figure 1 is that the samples are interrogated in situ i.e. from within the reactor and whilst there are changes as a function of temperature, there is no evidence that the same changes occur on exposure to an oxidising atmosphere). However, I think it is unfair to ask the authors to solve this problem as it is very challenging and beyond the scope of the study and that the real value of these results is that it serves as a demonstrator case. Besides which there has been a lot of effort to cross correlate the findings from SIM with other data (modelling, UV-Vis). I do think though that some comment regarding the time taken to acquire the SIM data and the time the samples were exposed to the air should be included lest others who embark on similar studies will end up laboring under the false pretense that all that is required is a SIM setup + few reacted samples. My PhD student who has performed similar studies in the past would argue that unless the atmosphere was controlled, then what you are measuring are not the species that are present during the reaction. Fortunately, she is not reviewing this manuscript.

Response: Thank the reviewer for valuable suggestion. The spent SAPO-34 zeolite samples were obtained by the following steps. MTO reaction catalyzed by calcined SAPO-34 zeolites sample was performed in fixed-bed reactor, and, when MTO reaction reached specific methanol conversion, the methanol feed was switched to nitrogen flow until temperature of fixed-bed reactor was rapidly cooled down below 473 K. Then spent SAPO-34 zeolite sample was taken out and rapidly charged to the sealed tube, and stored in the desiccator. During the shift of spent SAPO-34 zeolite samples, the contact of samples to oxidising atmosphere is inevitable, and the exposure time, including storage time, was within 24 hours. It was observed that below 473 K, the coke species have almost not been reacted with oxidising atmosphere in thermogravimetric analysis¹³. In addition, the study of operando UV-vis spectroscopy by Borodina et al.¹⁰ suggests that the UV spectra of spent H-SSZ-13 zeolite samples taken out from the reactor is similar to that of H-SSZ-13 samples directly obtained from in-situ reactions, and therefore, using

dissolution/extraction experiments to assign the characteristic adsorbed bands of UV spectra may be feasible. Nevertheless, we agree with the reviewer that the influence of offline measurement of catalyst samples by SIM needs to be carefully evaluated. At present, we are developing custom-made in-situ cell for MTO reaction which is expected to be used in SIM measurements. As this is a challenging work as also identified by the reviewer, it will take quite some time to obtain meaningful results. Meantime, since our current work focuses on the deep data approach, we will leave this interesting yet important topic for a future publication.

We have added above experimental steps in Page 27 in the revised manuscript.

Reference in response letter

1. Kalinin, S. V., Sumpster, B. G. & Archibald, R. K. Big-deep-smart data in imaging for guiding materials design. *Nat. Mater.* **14**, 973–980 (2015).
2. Hereijgers, B. P. C. *et al.* Product shape selectivity dominates the Methanol-to-Olefins (MTO) reaction over H-SAPO-34 catalysts. *J. Catal.* **264**, 77-87 (2009).
3. Yang, M. *et al.* A top-down approach to prepare silicoaluminophosphate molecular sieve nanocrystals with improved catalytic activity. *Chem. Commun.* **50**, 1845-1847 (2014).
4. Gao, M. *et al.* Direct quantification of surface barriers for mass transfer in nanoporous crystalline materials. *Commun. Chem.* **2**, 43-52 (2019).
5. Wang, C.-M., Wang, Y.-D. & Xie, Z.-K. Insights into the reaction mechanism of methanol-to-olefins conversion in HSAPO-34 from first principles: Are olefins themselves the dominating hydrocarbon pool species? *J. Catal.* **301**, 8-19 (2013).
6. Wang, S. *et al.* Polymethylbenzene or Alkene Cycle? Theoretical Study on Their Contribution to the Process of Methanol to Olefins over H-ZSM-5 Zeolite. *J. Phys. Chem. C* **119**, 28482-28498 (2015).
7. De Wispelaere, K., Hemelsoet, K., Waroquier, M. & Van Speybroeck, V. Complete low-barrier side-chain route for olefin formation during methanol conversion in H-SAPO-34. *J. Catal.* **305**, 76-80 (2013).
8. Hemelsoet, K. *et al.* Identification of Intermediates in Zeolite-Catalyzed Reactions by In Situ UV/Vis Microspectroscopy and a Complementary Set of Molecular Simulations. *Chem. Eur. J.* **19**, 16595-16606 (2013).
9. M. J. Frisch *et al.* *Gaussian 09*, Revision B.01 edn (Gaussian, Inc., Wallingford CT, 2016).
10. Borodina, E. *et al.* Influence of the Reaction Temperature on the Nature of the Active and Deactivating Species during Methanol to Olefins Conversion over H-SSZ-13. *ACS Catal.* **5**, 992-1003 (2015).
11. Omori, N. *et al.* Understanding the Dynamics of Fluorescence Emission during Zeolite Detemplation Using Time Resolved Photoluminescence Spectroscopy. *J. Phys. Chem. C* **124**, 531-543 (2020).
12. Van Speybroeck, V. *et al.* Mechanistic Studies on Chabazite-Type Methanol-to-Olefin Catalysts: Insights from Time-Resolved UV/Vis Microspectroscopy Combined with Theoretical Simulations. *ChemCatChem* **5**, 173-184 (2012).
13. Dai, W., Wu, G., Li, L., Guan, N. & Hunger, M. Mechanisms of the Deactivation of SAPO-34 Materials with Different Crystal Sizes Applied as MTO Catalysts. *ACS Catal.* **3**, 588-596 (2013).